# Effects of smoking on genome-wide DNA methylation profiles: A study of discordant and concordant monozygotic twin pairs

Jenny van Dongen[1,2,3]*, Gonneke Willemsen[1,2], BIOS Consortium, Eco JC de Geus[1,2], Dorret I Boomsma[1,2,3], Michael C Neale[4]

[1]Department of Biological Psychology, Vrije Universiteit Amsterdam, Amsterdam, Netherlands; [2]Amsterdam Public Health Research Institute, Amsterdam, Netherlands; [3]Amsterdam Reproduction and Development (AR&D) Research Institute, Amsterdam, Netherlands; [4]Virginia Institute for Psychiatric and Behavioral Genetics, Virginia Commonwealth University, Richmond, United States

*For correspondence:
j.van.dongen@vu.nl

Group author details:
BIOS Consortium See page 16

## Abstract

**Background:** Smoking-associated DNA methylation levels identified through epigenome-wide association studies (EWASs) are generally ascribed to smoking-reactive mechanisms, but the contribution of a shared genetic predisposition to smoking and DNA methylation levels is typically not accounted for.

**Methods:** We exploited a strong within-family design, that is, the discordant monozygotic twin design, to study reactiveness of DNA methylation in blood cells to smoking and reversibility of methylation patterns upon quitting smoking. Illumina HumanMethylation450 BeadChip data were available for 769 monozygotic twin pairs (mean age = 36 years, range = 18–78, 70% female), including pairs discordant or concordant for current or former smoking.

**Results:** In pairs discordant for current smoking, 13 differentially methylated CpGs were found between current smoking twins and their genetically identical co-twin who never smoked. Top sites include multiple CpGs in *CACNA1D* and *GNG12*, which encode subunits of a calcium voltage-gated channel and G protein, respectively. These proteins interact with the nicotinic acetylcholine receptor, suggesting that methylation levels at these CpGs might be reactive to nicotine exposure. All 13 CpGs have been previously associated with smoking in unrelated individuals and data from monozygotic pairs discordant for former smoking indicated that methylation patterns are to a large extent reversible upon smoking cessation. We further showed that differences in smoking level exposure for monozygotic twins who are both current smokers but differ in the number of cigarettes they smoke are reflected in their DNA methylation profiles.

**Conclusions:** In conclusion, by analysing data from monozygotic twins, we robustly demonstrate that DNA methylation level in human blood cells is reactive to cigarette smoking.

**Funding:** We acknowledge funding from the National Institute on Drug Abuse grant DA049867, the Netherlands Organization for Scientific Research (NWO): Biobanking and Biomolecular Research Infrastructure (BBMRI-NL, NWO 184.033.111) and the BBRMI-NL-financed BIOS Consortium (NWO 184.021.007), NWO Large Scale infrastructures X-Omics (184.034.019), Genotype/phenotype database for behaviour genetic and genetic epidemiological studies (ZonMw Middelgroot 911-09-032); Netherlands Twin Registry Repository: researching the interplay between genome and environment (NWO-Groot 480-15-001/674); the Avera Institute, Sioux Falls (USA), and the National Institutes of Health (NIH R01 HD042157-01A1, MH081802, Grand Opportunity grants 1RC2 MH089951 and

1RC2 MH089995); epigenetic data were generated at the Human Genomics Facility (HuGe-F) at ErasmusMC Rotterdam. Cotinine assaying was sponsored by the Neuroscience Campus Amsterdam. DIB acknowledges the Royal Netherlands Academy of Science Professor Award (PAH/6635).

## Editor's evaluation

This study presents valuable findings regarding how smoking can leave a lasting imprint on the human genome. The twin pairs study design is unique, and the methods applied by the authors are solid, providing an excellent starting point for large translational studies with rigorous laboratory approaches. This work will be of interest to geneticists and genetic epidemiologists.

## Introduction

Epigenome-wide association studies (EWASs) have identified robust differences in DNA methylation between smokers and non-smokers (*Gao et al., 2015*; *Heikkinen et al., 2022*). In a meta-analysis of blood-based DNA methylation studies (*N* = 15,907 individuals; the largest EWAS of smoking to date), 2623 CpG sites passed the Bonferroni threshold for genome-wide significance in a comparison of current and never smokers (*Joehanes et al., 2016*). Based on comparison with loci identified in large genome-wide association studies (GWASs), differentially methylated sites were significantly enriched in genes implicated in well-established smoking-associated diseases, such as cancer, cardiovascular disease, inflammatory disease, and lung disease, as well as in genes associated with schizophrenia and educational attainment (*Joehanes et al., 2016*). It has been hypothesized that smoking-induced methylation changes might also contribute to the addictive effect of smoking (*Zillich et al., 2022*).

Importantly, smoking-associated DNA methylation levels, as established in human EWA studies, may reflect different mechanisms. They may reflect causal effects of smoking on methylation, causal effects of methylation on smoking behaviour, methylation differences associated with epiphenomena of other exposures that correlate with smoking e.g. alcohol use (*Liu et al., 2018*) , or they may reflect a shared genetic predisposition to smoking and methylation level. To distinguish these different mechanisms require incisive study designs (*Vink et al., 2017*). Establishing whether methylation levels in smokers revert to levels of never smokers upon smoking cessation is a first step. A previous study of 2648 former smokers with cross-sectional methylation data from the Framingham Heart Study suggested that methylation levels at most CpGs return to the level of never smokers within 5 years after quitting smoking, but 36 CpGs were still differentially methylated in former smokers, who had quit smoking for 30 years (*Joehanes et al., 2016*). In the large EWAS meta-analysis of smoking (*Joehanes et al., 2016*), 185 CpGs were differentially methylated between former and never smokers (compared to 2623 between current and never smokers). In addition, differences between former and never smokers were smaller than between current and never smokers. Reversible DNA methylation patterns may suggest that DNA methylation is reactive to smoking. However, it is also possible that the different methylation level in current smokers reflects a higher genetic liability to smoking behaviour (that makes them more likely to initiate and keep smoking). Similarly, differences between former smokers and never smokers could reflect that smoking has caused a persistent methylation change but can also be driven by genetic factors.

In population-based studies, smoking cases and non-smoking individuals may differ on many aspects, including their genetic predisposition to smoking. On the other hand, monozygotic twins are genetically identical (except for de novo mutations, but these are rare [*Jonsson et al., 2021*; *Ouwens et al., 2018*]), share a womb, and are matched on sex, age, and childhood environment. They have been exposed to similar prenatal conditions, which may include second hand smoke from smoking mothers and others. Differences in prenatal environment of monozygotic twins due to for instance unequal vascular supply are also recognized (*Hall, 1996*; *Martin et al., 1997*), although it remains to be investigated to what extent the impact of prenatal smoke exposure might differ between monozygotic twins. Smoking discordant monozygotic twin pairs offer a unique opportunity to assess smoking-reactive DNA methylation patterns (*Leeuwen et al., 2007*; *Vink et al., 2017*). Despite the large number of previous population-based smoking EWASs, only one previous study compared genome-wide DNA methylation in smoking discordant monozygotic twin pairs (*Allione et al., 2015*). This study analysed whole-blood Illumina 450k array methylation data from 20 discordant pairs, and reported 22

**eLife digest** The genetic information of people who smoke present distinctive characteristics. In particular, previous research has revealed differences in patterns of DNA methylation, a type of chemical modification that helps cells switch certain genes on or off. However, most of these studies could not establish for sure whether these changes were caused by smoking, predisposed individuals to smoke, or were driven by underlying genetic variation in the DNA sequence itself.

To investigate this question, van Dongen et al. examined DNA methylation data from the blood cells of over 700 pairs of identical twins. These individuals share the exact same genetic information, making it possible to better evaluate the impact of lifestyle on DNA modifications.

The analyses identified differences in methylation at 13 DNA locations in pairs of twins where one was a current smoker and their sibling had never smoked. Two of the genes code for proteins involved in the response to nicotine, the primary addictive chemical in cigarette smoke. The differences were smaller if one of the twins had stopped smoking, suggesting that quitting can help to reverse some of these changes.

These findings confirm that DNA methylation in blood cells is influenced by cigarette smoke, which could help to better understand smoking-associated diseases. They also demonstrate how useful identical twins studies can be to identify methylation changes that are markers of lifestyle.

top loci, many of which had been previously associated with cigarette smoking in previous studies. However, following the correction for multiple testing, none of the differentially methylated loci were statistically significant, and this previous twin study did not examine reversibility of smoking effects, that is, where methylation status changes again following smoking cessation.

Here, we analyse unique data from a large cohort of monozygotic twin pairs. This cohort is sufficiently large to include current smoking discordant and concordant pairs, as well pairs discordant for former smoking (*Figure 1*). These groups allow identification of loci that are reactive to smoking, and examination of the extent to which effects are reversible upon quitting smoking. Monozygotic pairs in which both twins are current smokers, but who differ in quantity smoked, enable examination of the effects of smoking intensity. Finally, concordant pairs who never smoked allow assessment of the amount of DNA methylation variation at smoking-reactive loci that is due to non-genetic sources of variation other than smoking. In secondary enrichment analyses, we examined whether smoking-reactive methylation patterns are enriched (1) at loci detected in previous EWASs of other traits and exposures, (2) at loci detected in a previous large GWAS meta-analysis of smoking initiation (*Liu et al., 2019*) – these loci are presumed to have a causal effect on smoking behaviour, and (3) within Gene Ontology and Kegg pathways. Finally, we examined the relationship between DNA methylation and RNA transcript levels in blood for smoking-reactive loci.

## Methods
### Participants
In the Netherlands Twin Register (NTR), DNA methylation data are available for 3089 whole-blood samples from 3057 individuals in twin families, as described in detail previously (*van Dongen et al., 2016*). The samples were obtained from twins and family members, who participated in NTR longitudinal survey studies (*Ligthart et al., 2019*) and the NTR biobank project (*Willemsen et al., 2010*). In the current study, methylation data from monozygotic twin pairs were analysed. Among 768 monozygotic twin pairs with genome-wide methylation data and information on smoking and covariates, we identified the following discordant pairs: 53 discordant pairs, in which one twin was a current smoker at blood draw and the other never smoked, 72 discordant pairs, in which one twin was a former smoker at blood draw and the other never smoked, 66 discordant pairs of which one twin was a former smoker and the other a current smoker at blood draw. In addition, we identified the following concordant pairs: 83 twin pairs concordant for current smoking, 88 twin pairs concordant for former smoking, and 406 concordant twin pairs who never smoked. A flowchart is provided in *Figure 2*. Informed consent was obtained from all participants. The twin pairs were primarily of Dutch-European ancestry. For 753 of the 768 MZ pairs who are included in the current study, ancestry could be derived

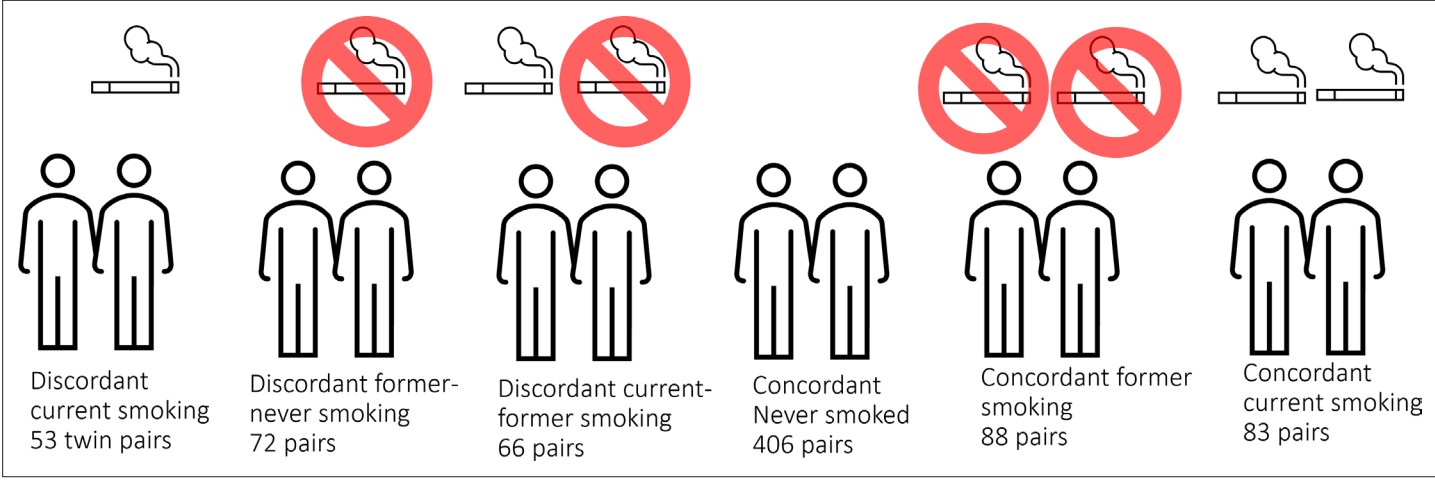

**Figure 1.** DNA methylation analysis in smoking discordant and smoking concordant monozygotic twin pairs. Blood DNA methylation profiles (Illumina 450k array) from six groups of monozygotic twin pairs were analysed.

from principal components (PCs) calculated from genome-wide Single Nucleotide Polymorphism (SNP) array data that were available for the twins (750 pairs) or for both of their parents (3 pairs). According to the genotype data PCs, 4.5% of the pairs classify as ancestry outliers.The study was approved by the Central Ethics Committee on Research Involving Human Subjects of the VU University Medical Centre, Amsterdam, an Institutional Review Board certified by the U.S. Office of Human Research Protections (IRB number IRB00002991 under Federal-wide Assurance – FWA00017598; IRB/institute code, NTR 03-180).

**Figure 2.** Study flowchart.

## Peripheral blood DNA methylation and cell counts

Genome-wide DNA methylation in whole blood was measured by the Human Genomics facility (HugeF) of ErasmusMC, the Netherlands (http://www.glimdna.org/). DNA methylation was assessed with the Infinium HumanMethylation450 BeadChip Kit (Illumina, San Diego, CA, USA). Genomic DNA (500 ng) from whole blood was bisulfite treated using the Zymo EZ DNA Methylation kit (Zymo Research Corp, Irvine, CA, USA), and 4 µl of bisulfite-converted DNA was measured on the Illumina 450k array (*Bibikova et al., 2011*) following the manufacturer's protocol. A custom pipeline for quality control and normalization of the methylation data was developed by the BIOS consortium. First, sample quality control was performed using MethylAid (*van Iterson et al., 2014*). Next, probe filtering was applied with DNAmArray (*Sinke et al., 2019*) to remove: ambiguously mapped probes (*Chen et al., 2013*), probes with a detection p-value >0.01, or bead number <3, or raw signal intensity of zero. After these probe filtering steps, probes and samples with a success rate <95% were removed. Next, the DNA methylation data were normalized using functional normalization (*Fortin et al., 2014*), as implemented in DNAmArray (*Sinke et al., 2019*) using the cohort-specific optimum number of control probe-based PCs. Probes containing an SNP, identified in a DNA sequencing project in the Dutch population (*The Genome of the Netherlands Consortium, 2014*), within the CpG site (at the C or G position) were excluded irrespective of minor allele frequency, and only autosomal probes were analysed, leading to a total number of 411,169 methylation sites. The following subtypes of white blood cells were counted in blood samples: neutrophils, lymphocytes, monocytes, eosinophils, and basophils (*Willemsen et al., 2010*).

## Smoking and other phenotypes

Information on smoking behaviour was obtained by interview during the home visit for blood collection as part of the NTR biobank project (2004–2008 and 2010–2011). The questions are included in *Supplementary file 1*. Participants were asked: 'Did you ever smoke?', with answer categories: (1) no, I never smoked, (2) I'm a former smoker, and (3) yes. Current smokers were asked how many years they smoked and how many cigarettes per day they smoked at present, while ex-smokers were asked how many years ago they quit, for how many years they smoked and how many cigarettes per day they smoked (note that the question on cigarettes per day to former smoker did not specify a particular time period, which may introduce variation in responses). Data were checked for consistencies and missing data were completed by linking this information to data from surveys filled out close to the time of biobanking within the longitudinal survey study of the NTR. More details on these checks are described in *Supplementary file 1*. Packyears were calculated as the (number of cigarettes smoked per day/20) × number of years smoked. Plasma cotinine level measurements have been described previously (*Bot et al., 2013*). Body mass index (BMI) was obtained at blood draw. Educational attainment was obtained in multiple longitudinal surveys and was defined as the highest completed level of education at the age of 25 or higher. It was classified on a 7-point scale: 1 = primary school only, 2 = lower vocational schooling, 3 = lower secondary schooling (general), 4 = intermediate vocational schooling, 5 = intermediate/higher secondary schooling (general), 6 = higher vocational schooling, 7 = university.

## Statistical analyses

### Overview and hypotheses

All analyses were performed in R (*R Development Core Team, 2013*). Analyses were performed in six groups of monozygotic twin pairs (*Figure 1*). To identify DNA methylation differences in smoking-discordant monozygotic twin pairs, we first compared the twin pairs, in which one twin had never smoked, and the other was a current smoker at the time of blood sampling. Second, to identify which of these DNA methylation differences might be reversible, we analysed data from (1) monozygotic pairs in which one twin had never smoked, and the other was a former smoker at the time of blood sampling, (2) from monozygotic pairs in which one twin was a current smoker, and the other was a former smoker at the time of blood sampling, and (3) from monozygotic pairs who were both former smokers. Third, to quantify within-pair methylation differences that occur by chance alone, we compared the within-pair differences monozygotic twins concordant for never having smoked. Forth, data from monozygotic twins concordant for current smoking were analysed to examine the effects of smoking intensity. Our hypotheses were as follows: (1) if DNA methylation level is reactive to cigarette

smoking, methylation differences will be present between smokers and non-smokers after ruling out genetic differences, that is in smoking-discordant monozygotic twin pairs, and these differences will be larger than in monozygotic pairs concordant for never smoking, (2) if DNA methylation patterns are reversible upon quitting smoking, methylation differences (ΔM) in monozygotic pairs will show the following pattern: ΔM discordant current-never > ΔM discordant current-former and ΔM discordant former-never > ΔM concordant never, (3) a correlation between time since quitting smoking and ΔM in pairs discordant for former smoking is consistent with a gradual reversibility of methylation levels upon quitting smoking, and (4) a correlation between ΔM and the difference in number of cigarettes smoked per day in smoking concordant pairs is consistent with smoking-reactive methylation patterns that show a dose–response relationship with amount of cigarettes smoked.

## Epigenome-wide association study

In the entire dataset of 3089 blood samples, we used linear regression analysis to correct the DNA methylation levels (β-values) for commonly used covariates (*van Rooij et al., 2019*), including HM450k array row, bisulphite plate (dummy-coding) and white blood cell percentages (% neutrophils, % monocytes, and % eosinophils). White blood cell percentages were included to account for variation in cellular composition between whole-blood samples. Lymphocyte percentage was not included in models because it was strongly correlated with neutrophil percentage ($r = -0.93$), and basophil percentage was not included because it showed little variation between subjects, with a large number of subjects having 0% of basophils. We did not adjust for sex and age, because monozygotic twins have the same sex and age. The residuals from this regression analysis were used in the within-pair EWAS analyses. Specifically, the residuals were used as input for paired *t*-tests to compare the methylation of the smoking twins with that of their non-smoking co-twins. Similarly, paired *t*-tests were applied to data from smoking concordant pairs. Statistical significance was assessed following stringent Bonferroni correction for the number of methylation sites tested ($\alpha = 0.05/411,169 = 1.2 \times 10^{-7}$). For each EWAS analysis, the R package Bacon was used to compute the Bayesian inflation factor (*van Iterson et al., 2017*). A previous power analysis for DNA methylation studies in discordant monozygotic twins indicated that with 50 discordant pairs, there is 80% power to detect methylation differences of 15% (at epigenome-wide significance; that is following multiple testing correction) (*Tsai and Bell, 2015*). Power quickly drops for smaller effect sizes; for example, with 50 discordant pairs, the power to detect a 10% methylation difference is 10% and the power to detect a methylation difference of 5% approaches alpha (*Tsai and Bell, 2015*). We tested for within-pair differences in demographics (e.g. BMI, educational attainment) and smoking characteristics (e.g. amount of cigarettes per day) with paired *t*-tests (continuous data) and Wilcoxon Signed Ranks tests (ordinal data) in R.

## Dose–response relationships

For significant CpGs from the EWAS of discordant monozygotic twin pairs, we examined dose–response relationships in smoking concordant pairs (both twins were current smokers) by correlating within-pair differences in DNA methylation with within-pair differences in smoking packyears and cigarettes per day. All correlations reported in this paper are Pearson correlations. Secondly, in twin pairs discordant for former smoking (one twin never smoked and the other one is a former smoker), we correlated and plotted within-pair differences in DNA methylation with the time since quitting smoking to assess the relationship between time since quitting smoking and reversal of methylation differences within monozygotic twin pairs.

## Enrichment analyses

We used the EWAS Toolkit from the EWAS atlas (*Li et al., 2019*) to perform enrichment analyses of Gene Ontology Terms, Kegg pathways, and previously associated traits among top sites from the EWAS in discordant monozygotic twin pairs (current versus never). With the trait enrichment tool of the EWAS analysis, we tested for enrichment of all traits (680) that were present in the atlas on 26 April 2022. Because the software requires a minimum of 20 input CpGs, we selected the top 20 CpGs from the EWAS in discordant monozygotic pairs for the enrichment analyses using the EWAS toolkit.

To study overlap of EWAS signal with genetic findings for smoking, we compared our EWAS results against GWAS results from the largest GWAS meta-analysis of smoking phenotypes. This is the meta-analysis of smoking initiation by the GWAS and Sequencing Consortium of Alcohol and Nicotine use

(GSCAN) (*Liu et al., 2019*). We obtained leave-one out meta-analysis results with NTR excluded. From the GWAS, we selected all SNPs with a p-value $<5.0 \times 10^{-8}$ and determined the distance of each Illumina 450k methylation site to each SNP. We then tested whether methylation sites within 1 Mb of genome-wide significant SNPs from the GWAS showed a stronger signal in the within-pair EWAS of smoking discordant monozygotic pairs compared to other genome-wide methylation sites, by regressing the EWAS test statistics on a variable (GWAS locus) indicating if the CpG is located within a 1 Mb window from SNPs associated with smoking initiation (1 = yes, 0 = no):

$$|t| = Intercept + \beta_{GWASlocus} * GWASlocus$$

where $|t|$ represents the absolute $t$-statistic from the paired $t$-test comparing within-pair methylation differences in smoking discordant pairs and $\beta_{GWASlocus}$ represents the estimate for $GWASlocus$, that is the change in the $t$-test statistic associated with a one-unit change in the variable $GWASlocus$ (e.g. being within 1 Mb of SNPs associated with smoking initiation). For each enrichment test, bootstrap standard erors were computed with 2000 bootstraps with the R-package 'simpleboot'.

## Gene expression

For significant CpGs from the EWAS of discordant monozygotic twin pairs (current versus never), we examined whether the DNA methylation was associated with gene expression levels in cis. To this end, we used an independent whole-blood RNA-sequencing dataset from the Biobank-based Integrative Omics Study (BIOS) consortium that did not include NTR, and tested associations between genome-wide CpGs and transcripts in cis (<250 kb), as described in detail previously (*the BIOS Consortium et al., 2017*). In short, methylation and expression levels in whole-blood samples ($n$ = 2101) were quantified with Illumina Infinium HumanMethylation450 BeadChip arrays and with RNA-seq (2 × 50 bp paired-end, Hiseq2000, >15 M read pairs per sample). For each target CpG (epigenome-wide significant differentially methylated positions [DMPs]), we identified transcripts in cis (<250 kb), for which methylation levels were significantly associated with gene expression levels at the experiment-wide threshold applied by this study (False Discovery Rate (FDR) <5.0%), after regressing out methylation Quantitative Trait Locus (mQTL) and expression Quantitative Trait Locus (eQTL) effects. We also examined whether significant CpGs from the EWAS of discordant monozygotic twin pairs mapped to genes that were previously reported to be differentially expressed in monozygotic pairs of which one twin never smoked, and the other was a current smoker at the time of blood sampling (based on Affymetrix U219 array data; $n$ = 56 pairs; note: the 53 discordant pairs included in the current study of DNA methylation are a subset of the 56 discordant pairs included in the study of gene expression) (*Vink et al., 2017*).

## Results

Descriptives of the smoking-discordant and concordant monozygotic twin pairs are given in *Table 1*. In twin pairs discordant for current smoking status (i.e. one twin a current smoker at the time of blood sampling and the other never initiated regular smoking, $N$ = 53 pairs, mean age = 33 years), the smoking twin on average smoked 8.9 cigarettes per day at the time of blood sampling, and had an average smoking history equivalent to 6.8 packyears. The EWAS analysis in pairs discordant for current smoking status identified 13 epigenome-wide significant (p < $1.20 \times 10^{-7}$) DMPs (*Figure 3a*). Genome-wide test statistics were not inflated (*Supplementary file 2*). Absolute differences in methylation ranged from 2.5% to 13% (0.025–0.13 on the methylation $\beta$-value scale), with a mean of 5.4% (*Table 2*). Eight of the 13 CpGs (61.5%) showed lower methylation in the current smoking twins compared to their non-smoking twins. Pair-level methylation $\beta$-values are shown in *Figure 3—figure supplement 1* and illustrate large consistency in the direction of effect. For example, at top CpG site cg05575921, for 51 out of the 53 pairs, the smoking twin had a lower methylation level than the non-smoking twin. At 11 of the 13 CpGs, the methylation difference in smoking discordant monozygotic twin pairs was smaller (on average 19.0%, range = 2.2–37.5%) compared to the methylation difference reported previously in an EWAS meta-analysis of smoking (*Joehanes et al., 2016*). At two CpGs, the methylation difference in smoking discordant monozygotic twins was larger (on average 24.6%).

In twin pairs discordant for former smoking ($N$ = 72 pairs, mean age = 41 years), the twins, who used to smoke, had quit smoking on average 14 years ago (standard deviation [SD] = 11.4,

**Table 1.** Descriptive statistics for smoking discordant and concordant monozygotic twin pairs.

| | Discordant current/never (53 pairs) | | | | Discordant former/never (72 pairs) | | | | Discordant current/former (66 pairs) | | | | Concordant current (83 pairs) | | | | Concordant never (406 pairs) | | | | Concordant former (88 pairs) | | | |
|---|---|---|---|---|---|---|---|---|---|---|---|---|---|---|---|---|---|---|---|---|---|---|---|---|
| | Current smoker | Never-smoker | Mean diff | p-value | Former smoker | Never-smoker | Mean diff | p-value | Current smoker | Former smoker | Mean diff | p-value | Twin 1 | Twin 2 | Mean diff | p-value | Twin 1 | Twin 2 | Mean diff | p-value | Twin 1 | Twin 2 | Mean diff | p-value |
| % Female pairs | 60.4% | 60.4% | n.a. | n.a. | 77.80% | 77.80% | n.a. | n.a. | 69.7% | 69.7% | n.a. | n.a. | 61.4% | 61.4% | n.a. | n.a. | 73.6% | 73.6% | n.a. | n.a. | 64.8% | 64.8% | n.a. | n.a. |
| Age at blood sampling, mean (SD) | 33.1 (8.0) | 33.0 (7.9) | 0.10 | 0.34 | 41.4 (13.2) | 41.4 (13.1) | 0.02 | 0.83 | 42.2 (12.6) | 42.2 (12.5) | −0.06 | 0.45 | 33.8 (10.3) | 33.9 (10.5) | −0.12 | 0.10 | 33.1 (11.3) | 33.0 (11.2) | 0.06 | 0.08 | 45.2 (13.4) | 45.2 (13.4) | 0.09 | 0.29 |
| Cigarettes per day at blood sampling, mean (SD), N missings | 8.9 (6.4), 6 | n.a. | n.a. | n.a. | n.a. | n.a. | n.a. | n.a. | 11.9 (7.2), 9 | n.a. | n.a. | n.a. | 11.1 (7.0), 2 | 10.9 (6.9), 1 | 0.00 | 1.00 | n.a. | n.a. | n.a. | n.a. | n.a. | n.a. | n.a. | n.a. |
| Packyears, mean (SD), N missings | 6.8 (7.0), 13 | n.a. | n.a. | n.a. | 5.9 (11.1), 15 | n.a. | n.a. | n.a. | 13.6 (13.2), 9 | 9.3 (8.7), 10 | 3.9 | 0.05 | 9.7 (9.3), 10 | 8.3 (7.6), 9 | 0.22 | 0.82 | n.a. | n.a. | n.a. | n.a. | 10.6 (11.5), 7 | 9.8 (10.4), 11 | 0.78 | 0.55 |
| Years since quitting smoking, mean (SD), N missings | n.a. | n.a. | n.a. | n.a. | 13.5 (11.4), 9 | n.a. | n.a. | n.a. | n.a. | 9.0 (10.2), 2 | n.a. | n.a. | n.a. | n.a. | n.a. | n.a. | n.a. | n.a. | n.a. | n.a. | 11.9 (9.1), 8 | 13.6 (11.8), 7 | −1.62 | 0.20 |
| Plasma cotinine level, mean (SD), N missings* | 222 (197.5), 1 | 1.8 (2.6), 28 | 261.1 | 1.6 × 10^-5 | 1.4 (1.8), 49 | 0.9 (1.0), 52 | 0.62 | 0.43 | 286.7 (330.5), 2 | 19.2 (70.0), 28 | 293.8 | 2.5 × 10^-6 | 267 (290.9), 3 | 279 (308.1), 3 | −8.7 | 0.78 | 1.3 (9.7), 274 | 0.5 (0.8), 260 | 0.07 | 0.50 | 55.8 (222.2), 61 | 6 (16.2), 64 | 83.3 | 0.26 |
| Educational Attainment, N (%) | | | | 0.04 | | | | 0.97 | | | | 0.34 | | | | 0.76 | | | | 0.76 | | | | 0.83 |
| N missing | 18 | 11 | | | 13 | 13 | | | 14 | 14 | | | 28 | 34 | | | 105 | 108 | | | 14 | 20 | | |
| 1. Primary school only | 0 (0%) | 1 (2.3%) | | | 0 (0%) | 1 (1.7%) | | | 2 (3.8%) | 5 (9.6%) | | | 1 (1.8%) | 2 (4.1%) | | | 3 (1.0%) | 4 (1.3%) | | | 6 (8.1%) | 4 (5.9%) | | |
| 2. Lower vocational schooling | 1 (2.9%) | 0 (0%) | | | 7 (11.9%) | 6 (10.2%) | | | 10 (19.2%) | 4 (7.7%) | | | 4 (7.3%) | 1 (2.0%) | | | 8 (2.7%) | 10 (3.4%) | | | 2 (2.7%) | 10 (14.7%) | | |
| 3. Lower secondary schooling (general) | 2 (5.7%) | 2 (4.8%) | | | 11 (18.6%) | 8 (13.6%) | | | 7 (13.5%) | 4 (7.7%) | | | 5 (9.1%) | 7 (14.3%) | | | 11 (3.7%) | 11 (3.7%) | | | 10 (13.5%) | 6 (8.8%) | | |
| 4. Intermediate vocational schooling | 12 (34.3%) | 15 (35.7%) | | | 13 (23.7%) | 19 (32.2%) | | | 8 (15.4%) | 13 (25.0%) | | | 16 (29.1%) | 16 (32.7%) | | | 81 (26.9%) | 85 (28.5%) | | | 26 (35.1%) | 17 (25.0%) | | |
| 5. Intermediate/higher secondary schooling (general) | 2 (5.7%) | 1 (2.4%) | | | 3 (6.8%) | 3 (5.1%) | | | 2 (3.8%) | 2 (3.8%) | | | 4 (7.3%) | 6 (12.2%) | | | 15 (5.0%) | 17 (5.7%) | | | 3 (4.1%) | 2 (2.9%) | | |
| 6. Higher vocational schooling | 14 (40.0%) | 14 (33.3%) | | | 14 (22.0%) | 11 (18.6%) | | | 15 (28.8%) | 16 (30.8%) | | | 19 (34.5%) | 11 (22.4%) | | | 97 (32.2%) | 81 (27.2%) | | | 17 (23.0%) | 17 (25.0%) | | |
| 7. University | 4 (11.4%) | 9 (21.4%) | | | 10 (16.9%) | 11 (18.6%) | | | 8 (15.4%) | 8 (15.4%) | | | 6 (10.9%) | 6 (12.2%) | | | 86 (28.6%) | 90 (30.2%) | | | 10 (13.5%) | 12 (17.6%) | | |
| BMI, mean (SD), N missings | 23.8 (3.8), n.a. | 24.0 (3.4), n.a. | −0.17 | 0.74 | 25.0 (3.6), n.a. | 25.1 (4.2), n.a. | −0.13 | 0.75 | 23.7 (3.1) | 25.2 (4.2) | −1.47 | 3.4 × 10^-4 | 23.6 (3.5), n.a. | 23.1 (3.2), n.a. | 0.47 | 0.06 | 23.8 (3.8),4 | 23.6 (3.4), 2 | 0.27 | 0.03 | 25.6 (4.0), n.a. | 24.9 (3.7), n.a. | 0.63 | 0.02 |
| Percentage monocytes, mean (SD), N missings | 8.0 (2.3), 0 | 8.5 (2.4), 0 | −0.44 | 0.19 | 8.6 (2.0), 0 | 8.3 (1.8), 0 | 0.29 | 0.19 | 8.6 (1.9), 0 | 9.2 (3.1), 0 | −0.57 | 0.14 | 8.3 (2.1), 0 | 8.1 (1.9), 0 | 0.20 | 0.36 | 8.5 (2.0), 0 | 8.5 (2.2), 0 | 0.03 | 0.75 | 8.4 (2.4), 0 | 8.5 (2.4), 0 | −0.06 | 0.75 |
| Percentage lymphocytes, mean (SD), N missings | 35.0 (8.9), 0 | 35.9 (10.0), 0 | −0.94 | 0.50 | 33.6 (8.5), 0 | 34.0 (8.7) | −0.37 | 0.77 | 35.8 (8.2),0 | 35.6 (8.5), 0 | 0.25 | 0.84 | 33.7 (8.3), 0 | 34.1 (8.3), 0 | −0.44 | 0.67 | 36.3 (8.4), 0 | 36.2 (8.4), 0 | 0.04 | 0.92 | 35.0 (7.7), 0 | 34.1 (8.4), 0 | 0.95 | 0.26 |
| Percentage neutrophils, mean (SD), N missings | 53.4 (9.5), 0 | 52.1 (9.8), 0 | 1.34 | 0.38 | 54.4 (9.1), 0 | 54.5 (9.1), 0 | −0.08 | 0.95 | 51.6 (9.0),0 | 51.7 (8.9), 0 | −0.06 | 0.96 | 53.7 (8.9), 0 | 53.7 (9.3), 0 | 0.03 | 0.98 | 51.8 (8.7), 0 | 51.9 (9.3), 0 | −0.08 | 0.86 | 52.8 (8.2), 0 | 53.7 (8.4), 0 | −0.84 | 0.35 |
| Percentage eosinophils, mean (SD), N missings | 3.1 (2.5), 0 | 3.1 (2.1), 0 | 0.05 | 0.91 | 3.1 (1.9),0 | 2.9 (2.0), 0 | 0.15 | 0.53 | 3.3 (1.9),0 | 3.1 (1.7), 0 | 0.21 | 0.33 | 3.4 (2.2), 0 | 3.4 (1.8), 0 | −0.03 | 0.91 | 2.9 (1.8), 0 | 2.9 (1.9), 0 | −0.04 | 0.66 | 3.1 (1.9), 0 | 3.2 (2.4), 0 | −0.06 | 0.77 |
| Percentage basophils, mean (SD), N missings | 0.5 (0.7), 0 | 0.5 (0.7), 0 | −0.01 | 0.96 | 0.3 (0.3), 0 | 0.4 (0.5), 0 | −0.02 | 0.76 | 0.6 (0.9),0 | 0.4 (0.4), 0 | 0.18 | 0.12 | 0.9 (3.1), 0 | 0.6 (1.1), 0 | 0.25 | 0.49 | 0.5 (0.7), 0 | 0.4 (0.7), 0 | 0.06 | 0.21 | 0.6 (1.1), 0 | 0.6 (0.9), 0 | −0.01 | 0.97 |

*Missing values include values that are below the detection limit. BMI = body mass index.

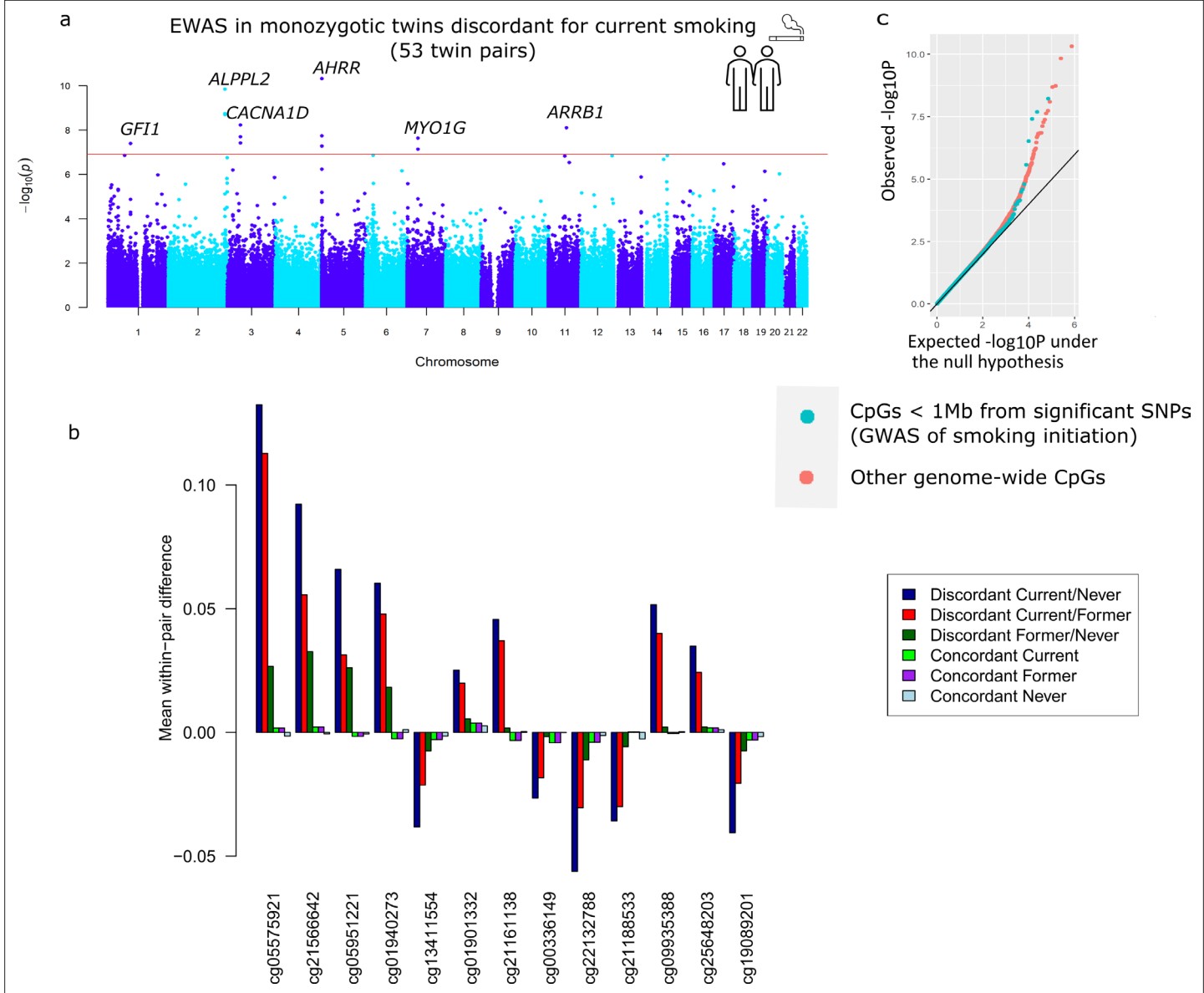

**Figure 3.** Top differentially methylated loci identified in monozygotic twin pairs discordant for current smoking. (**a**) Manhattan plot of the epigenome-wide association study (EWAS) in 53 smoking discordant monozygotic twin pairs (current versus never). The red horizontal line denotes the epigenome-wide significance threshold (Bonferroni correction) and 13 CpGs with significant differences are highlighted. (**b**) Mean within-pair differences in monozygotic twin pairs at the 13 CpGs that were epigenome-wide significant in smoking discordant monozygotic pairs. Mean within-pair differences of the residuals obtained after correction of methylation *β*-values for covariates are shown for 53 monozygotic pairs discordant for current/never smoking, 66 monozygotic pairs discordant for current/former smoking, 72 monozygotic pairs discordant for former/never smoking, 83 concordant current smoking monozygotic pairs, 88 concordant former smoking monozygotic pairs, and 406 concordant never smoking monozygotic pairs. (**c**) QQ-plot showing p-values from the EWAS in 53 smoking discordant monozygotic twin pairs (current versus never). P-values for CpGs located nearby significant SNPs from the genome-wide association study (GWAS) of smoking initiation are plotted in blue and all other genome-wide CpGs are plotted in orange.

The online version of this article includes the following figure supplement(s) for figure 3:

**Figure supplement 1.** DNA methylation levels in current/never smoking discordant monozygotic twin pairs.

range = 0.04–50 years), while the other twins had never initiated regular smoking. In this group, no epigenome-wide significant DMPs were identified, and within-pair differences at the 13 significant DMPs identified in the previous analysis were diminished (average reduction: 81%, range = 61–96%; *Figure 3b*, *Table 2*). By contrast, in twin pairs of which one twin was a current smoker at blood draw and the co-twin had quit smoking (on average 9 years, ago, SD = 10.2, range = 0.02–40 years, *N* =

**Table 2.** Epigenome-wide significant differentially methylated CpGs in monozygotic pairs discordant for current smoking status.

| IlmnID | CHR | MAPINFO | Gene* | Nearest gene | Current smoking discordant pairs | | | | | Former smoking discordant pairs (former/never) | | | | |
|---|---|---|---|---|---|---|---|---|---|---|---|---|---|---|
| | | | | | Mean diff | p-value | 95conf_L | 95conf_H | T-Statistic | Mean diff | p-value | 95conf_L | 95conf_H | T-Statistic |
| cg05575921 | 5 | 373378 | AHRR | AHRR | 0.132 | $4.9 \times 10^{-11}$ | 0.100 | 0.165 | 8.265 | 0.027 | $3.3 \times 10^{-4}$ | 0.013 | 0.041 | 3.778 |
| cg21566642 | 2 | 233284661 | | ALPPL2 | 0.092 | $1.5 \times 10^{-10}$ | 0.069 | 0.115 | 7.960 | 0.033 | $3.2 \times 10^{-6}$ | 0.020 | 0.046 | 5.067 |
| cg05951221 | 2 | 233284402 | ALPPL2 | ALPPL2 | 0.066 | $1.8 \times 10^{-9}$ | 0.048 | 0.084 | 7.270 | 0.026 | $4.6 \times 10^{-6}$ | 0.016 | 0.037 | 4.964 |
| cg01940273 | 2 | 233284934 | ALPPL2 | ALPPL2 | 0.060 | $2.1 \times 10^{-9}$ | 0.044 | 0.077 | 7.240 | 0.018 | $1.3 \times 10^{-4}$ | 0.009 | 0.027 | 4.052 |
| cg13411554 | 3 | 53700276 | CACNA1D | CACNA1D | -0.038 | $6.0 \times 10^{-9}$ | -0.049 | -0.027 | -6.947 | -0.007 | 0.10 | -0.016 | 0.002 | -1.655 |
| cg01901332 | 11 | 75031054 | ARRB1 | ARRB1 | 0.025 | $8.0 \times 10^{-9}$ | 0.018 | 0.033 | 6.868 | 0.006 | 0.16 | -0.002 | 0.013 | 1.425 |
| cg21161138 | 5 | 399360 | AHRR | AHRR | 0.046 | $1.9 \times 10^{-8}$ | 0.032 | 0.059 | 6.642 | 0.002 | 0.64 | -0.006 | 0.009 | 0.466 |
| cg00336149 | 3 | 53700195 | CACNA1D | CACNA1D | -0.027 | $2.0 \times 10^{-8}$ | -0.035 | -0.019 | -6.615 | -0.002 | 0.60 | -0.008 | 0.005 | -0.524 |
| cg22132788 | 7 | 45002486 | MYO1G | MYO1G | -0.056 | $2.4 \times 10^{-8}$ | -0.073 | -0.039 | -6.596 | -0.011 | $4.5 \times 10^{-3}$ | -0.019 | -0.004 | -2.930 |
| cg21188533 | 3 | 53700263 | CACNA1D | CACNA1D | -0.036 | $3.9 \times 10^{-8}$ | -0.047 | -0.025 | -6.437 | -0.006 | 0.24 | -0.015 | 0.004 | -1.196 |
| cg09935388 | 1 | 92947588 | GFI1 | GFI1 | 0.052 | $4.1 \times 10^{-8}$ | 0.035 | 0.068 | 6.423 | 0.002 | 0.61 | -0.006 | 0.010 | 0.519 |
| cg25648203 | 5 | 395444 | AHRR | AHRR | 0.035 | $5.3 \times 10^{-8}$ | 0.024 | 0.046 | 6.353 | 0.002 | 0.48 | -0.004 | 0.008 | 0.710 |
| cg19089201 | 7 | 45002287 | MYO1G | MYO1G | -0.040 | $7.5 \times 10^{-8}$ | -0.053 | -0.028 | -6.260 | -0.007 | 0.13 | -0.017 | 0.002 | -1.529 |

Coordinates are given based on genome build 37. Mean differences represent non-smoking twin minus smoking-twin (hence positive values indicate a higher methylation level in non-smoking twins). The table shows the 13 epigenome-wide significant CpGs from the within-pair EWAS in 53 discordant monozygotic twin pairs (current versus never smokers). Results from the comparison within 72 monozygotic pairs discordant for former smoking are also shown.

*CpGs without a gene name are located in intergenic regions. 95conf_L = 95% confidence interval lower bound, 95conf_H = 95% confidence interval upper bound.

66 pairs), the reduction of within-pair differences at the 13 top CpGs was much smaller (on average, 31%, range 15–52%; *Figure 3b*, *Supplementary file 3*), and 5 of the 13 DMPs identified by comparing current and never smoking twins were also epigenome-wide significant in this group. Furthermore, five additional epigenome-wide CpGs were identified in current/former smoking discordant pairs (*Supplementary file 4*). *Figure 3b* illustrates the pattern of within-pair differences at the 13 top DMPs identified in current/never discordant monozygotic pairs: largest differences in current/never smoking discordant pairs, smaller differences in former/never discordant pair, and current/former discordant pairs are intermediate. Differences are smallest within smoking concordant pairs. This pattern is in line with smoking-associated methylation patterns in blood cells being to a large extent reversible upon quitting smoking.

Distributions of within-pair differences in smoking discordant and concordant pairs for the top 1000 CpGs of the EWAS in discordant pairs are shown in *Figure 4a*. The distributions illustrate that differences are largest, as expected, within monozygotic twin pairs discordant for current smoking (current/never smoking pairs), followed by discordant current/former smoking discordant pairs, followed by former/never smoking discordant monozygotic twin pairs. Monozygotic pairs concordant for current smoking also show notable within-pair differences at these CpGs that are substantially larger compared to monozygotic pairs concordant for never smoking (*Figure 4a*). This could be explained by within-pair differences in the number of cigarettes smoked by monozygotic twins who were concordant for current smoking. The twin correlations in current smoking monozygotic twin pairs were $r = 0.50$, $p = 2.2 \times 10^{-6}$ for cigarettes per day (*Figure 4b*)**,** $r = 0.43$, $p = 3.2 \times 10^{-4}$ for packyears, and $r = 0.58$, $p = 1.6 \times 10^{-8}$, for plasma cotinine levels, respectively. Within-pair differences in DNA methylation at the 13 top CpGs correlated with within-pair differences in the number of cigarettes smoked per day (mean absolute $r = 0.38$, range [for different CpGs]: −0.56 to 0.41; *Table 3*, *Figure 4c*),with within-pair differences in packyears (mean absolute $r = 0.46$, range: −0.65 to 0.42; *Table 3*), but did not correlate strongly with within-pair differences in plasma cotinine level (mean absolute $r = 0.14$, range: −0.23 to 0.28, *Table 3*). In twin pairs discordant for former smoking, within-pair differences in DNA methylation at the 13 top CpGs were weakly correlated with time since quitting smoking (mean $r = −0.11$, range = −0.28 to 0.05, *Supplementary file 5*). Based on scatterplots of the within-pair methylation differences against time since quitting smoking (*Figure 4d*), we hypothesized that the lack of a strong correlation with time since quitting smoking might be explained by most of the reversal taking place within the first years after quitting smoking. We therefore repeated the analysis restricting to those pairs of which the smoking twin had quit smoking less than 5 years ago ($N = 15$ pairs). In this group, within-pair differences in DNA methylation at the 13 top CpGs were on average more strongly correlated with time since quitting smoking (mean $r = −0.16$, range = −0.48 to 0.23) but the sample size was greatly reduced and correlations were non-significant.

All 13 differentially methylated CpGs identified in current smoking discordant pairs and all 10 CpGs identified in former smoking discordant pairs have been previously associated with smoking. To study the overlap of methylation differences in smoking discordant twin pairs with loci that have a causal effect on smoking, we considered the largest GWAS meta-analysis of smoking phenotypes, the meta-analysis of smoking initiation by the GWAS and Sequencing Consortium of Alcohol and Nicotine use (GSCAN) (*Liu et al., 2019*). Three of the 13 epigenome-wide significant DMPs detected in smoking discordant monozygotic pairs (cg13411554, cg00336149, and cg21188533 in *CACNA1D*) are located within 1 Mb of a GWAS locus associated with smoking initiation. The methylation sites within 1 Mb of genome-wide significant SNPs from the GWAS overall did not show a stronger signal in the within-pair EWAS of smoking discordant monozygotic pairs compared to other genome-wide methylation sites ($\beta = −0.002$, se = 0.004, p = 0.56, *Figure 3c*).

We tested for enrichment of methylation sites previously associated with 680 traits reported in the EWAS atlas (*Li et al., 2019*), among the top differentially methylated loci in smoking discordant pairs, which showed strong enrichment of smoking-related traits (*Supplementary file 6*). Enrichment analysis based on Kegg pathways showed one significantly enriched pathway; Dopaminergic Synapse (hsa04728; *Supplementary file 7*), with three of the top differentially methylated loci in smoking discordant monozygotic pairs mapping to this pathway: *CACNA1D*, *GNG12*, and *ARRB1*. No significant enrichment was seen in GO pathways after multiple testing correction (*Supplementary file 8*).

To examine potential functional consequences of top DMPs, we used previously published data on whole-blood DNA methylation and RNA-sequencing ($n = 2101$ samples). At 4 of the 13 CpGs, DNA

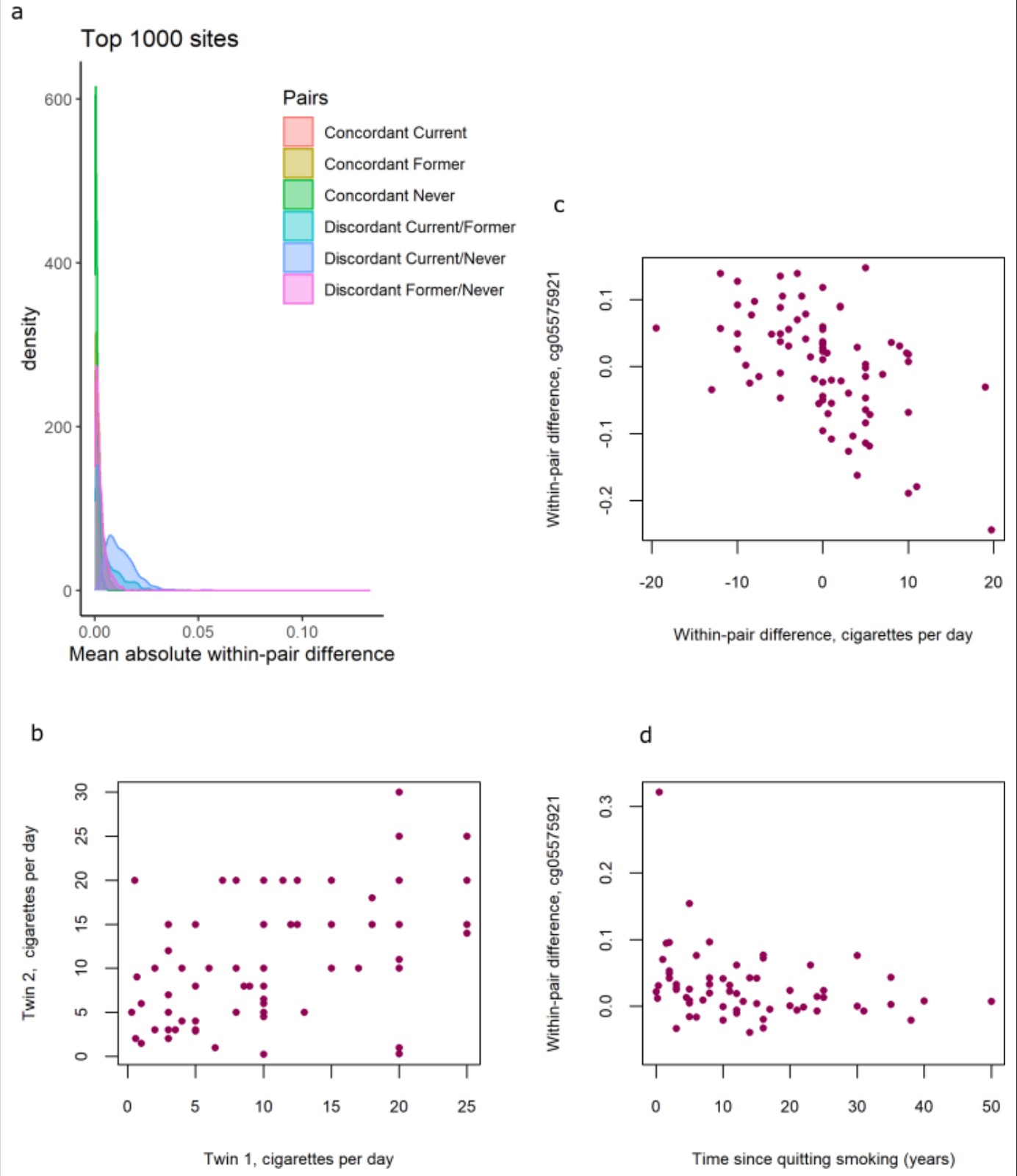

**Figure 4.** DNA methylation differences in smoking discordant and smoking concordant pairs. (**a**) Distributions of the mean absolute within-pair differences in discordant and concordant pairs at the top 1000 CpGs with the lowest p-value from the epigenome-wide association study (EWAS) in discordant monozygotic pairs (current versus never smokers). (**b**) Scatterplot of cigarettes smoked per day in 80 concordant current smoking monozygotic pairs with complete data. (**c**) Scatterplot of within-pair differences in cigarettes smoked per day versus DNA methylation at cg05575921

*Figure 4 continued on next page*

Figure 4 continued

(*AHRR*) in 80 concordant current smoking monozygotic pairs with complete data. (**d**) Scatterplot of within-pair differences in DNA methylation at cg05575921 (*AHRR*) versus time since quitting smoking (years) in 63 pairs discordant for former smoking.

methylation level in blood was associated with the expression level of nearby genes (*Table 4*). At three CpGs, a higher methylation level correlated with lower expression level. None of the 13 CpGs overlapped with six genes that were differentially expressed in monozygotic pairs discordant for current smoking (*Vink et al., 2017*).

## Discussion

Previous EWASs have identified robust differences in DNA methylation between smokers and non-smokers at a number of loci. These differences may reflect true smoking-reactive DNA methylation patterns, but can also be driven by (genetic) confounding or reverse causation. We exploited a strong within-family design, that is, the discordant monozygotic twin design (*Bell and Spector, 2012*), to identify smoking-reactive loci. By analysing whole-blood genome-wide DNA methylation patterns in 53 monozygotic pairs discordant for current smoking, we found 13 CpGs with a difference in methylation level between the current smoking twin and the twin who never smoked. All 13 CpGs have been previously associated with smoking in unrelated individuals and in line with previous studies that compared unrelated smokers and controls (*Joehanes et al., 2016*), our data from monozygotic pairs discordant for former smoking also indicate that methylation patterns are to a large extent reversible upon smoking cessation. We further showed that differences in smoking level exposure for monozygotic twins who are both current smokers but differ in the number of cigarettes they smoke are reflected in their DNA methylation profiles.

The strongest smoking-associated loci typically detected in human blood EWAS are genes involved in detoxification pathways of aromatic hydrocarbons, such as *AHRR* and *CYP1A1* (*Gao et al., 2015*), of which *AHRR* was also present among the top differentially methylated loci in our analysis of discordant twin pairs. Mainstream tobacco smoke is a mixture of thousands of chemicals (*Rodgman and Perfetti, 2008*). Although the effects of many of the compounds present in cigarette smoke are unknown, several mechanisms have been described through which cigarette smoking may affect global or gene-specific DNA methylation levels. These include DNA damage induced by certain compounds such as arsenic, chromium, formaldehyde, polycyclic aromatic hydrocarbons, and nitrosamines that all cause double-stranded breaks (*Smith and Hansch, 2000*) (which causes increased methylation near repaired DNA) (*Mortusewicz et al., 2005*; *Cuozzo et al., 2007*), hypoxia induced by carbon monoxide (*Olson, 1984*) (causing global CpG island demethylation by disrupting methyl donor availability), and modulation of the expression level or activity of DNA-binding proteins, such as transcription factors (*Lee and Pausova, 2013*). Nicotine, presumed to be the major addictive compound in cigarette smoke (although other putative addictive compounds have also been described [*Talhout et al., 2011*]), has gene regulatory effects. Binding of nicotine to nicotinic acetylcholine receptors causes downstream activation of cAMP response element-binding protein, which is a key transcription factor for many genes (*Shen and Yakel, 2009*). In mouse brain, nicotine downregulates the DNA methyl transferase gene *Dnmt* (*Satta et al., 2008*). Previous EWAS studies based on blood cotinine levels, as a biomarker for nicotine exposure, and based on a polygenic scores for nicotine metabolism, reported differentially methylated CpGs that largely overlap with CpGs found in EWAS of smoking status (*Gupta et al., 2019*; *Lee et al., 2016*). Furthermore, E-cigarette-based nicotine exposure of mice has been shown to cause DNA methylation changes in white blood cells (*Peng et al., 2022*).

Importantly, effects of smoking on DNA methylation in brain cells have been hypothesized to contribute to addiction (*Zillich et al., 2022*), but it is largely unknown to what extent addiction-related DNA methylation dynamics are captured in other tissues such as blood. Nicotinic receptors are especially abundant in the central and peripheral nervous system, but are also present in other tissues. In peripheral blood, nicotinic receptors are present on lymphocytes and polymorphonuclear cells (*Benhammou et al., 2000*), suggesting that EWA studies performed on blood cells might capture nicotine-reactive methylation patterns. Interesting in this regard is our finding that among the top differentially methylated CpGs in smoking discordant pairs are multiple CpGs in *CACNA1D* and *GNG12*, which encode subunits of a calcium voltage-gated channel and G protein, respectively;

**Table 3.** Correlations of within-pair differences in DNA methylation with within-pair differences in cigarettes per day and packyears in 83 concordant current smoking monozygotic pairs.

| cgid | CHR | Position | Gene | Nearest gene | Cigarettes per day | | Packyears | |
|---|---|---|---|---|---|---|---|---|
| | | | | | r | p-value | r | p-value |
| cg05575921 | 5 | 373378 | AHRR | AHRR | −0.52 | $5.9 \times 10^{-7}$ | −0.55 | $1.2 \times 10^{-6}$ | −0.08 | 0.50 |
| cg21566642 | 2 | 233284661 | | ALPPL2 | −0.49 | $4.7 \times 10^{-6}$ | −0.56 | $8.1 \times 10^{-7}$ | −0.19 | 0.10 |
| cg05951221 | 2 | 233284402 | ALPPL2 | ALPPL2 | −0.44 | $4.7 \times 10^{-5}$ | −0.56 | $9.0 \times 10^{-7}$ | −0.18 | 0.12 |
| cg01940273 | 2 | 233284934 | ALPPL2 | ALPPL2 | −0.56 | $7.0 \times 10^{-8}$ | −0.65 | $2.0 \times 10^{-9}$ | −0.23 | 0.04 |
| cg13411554 | 3 | 53700276 | CACNA1D | CACNA1D | 0.27 | $1.4 \times 10^{-2}$ | 0.32 | $7.4 \times 10^{-3}$ | 0.20 | 0.08 |
| cg01901332 | 11 | 75031054 | ARRB1 | ARRB1 | −0.23 | $3.8 \times 10^{-2}$ | −0.34 | $5.5 \times 10^{-3}$ | −0.04 | 0.71 |
| cg21161138 | 5 | 399360 | AHRR | AHRR | −0.52 | $8.4 \times 10^{-7}$ | −0.52 | $7.4 \times 10^{-6}$ | −0.05 | 0.65 |
| cg00336149 | 3 | 53700195 | CACNA1D | CACNA1D | 0.36 | $1.0 \times 10^{-3}$ | 0.42 | $3.5 \times 10^{-4}$ | 0.16 | 0.16 |
| cg22132788 | 7 | 45002486 | MYO1G | MYO1G | 0.41 | $2.1 \times 10^{-4}$ | 0.41 | $8.4 \times 10^{-4}$ | 0.14 | 0.23 |
| cg21188533 | 3 | 53700263 | CACNA1D | CACNA1D | 0.32 | $4.1 \times 10^{-3}$ | 0.38 | $1.4 \times 10^{-3}$ | 0.27 | 0.01 |
| cg09935388 | 1 | 92947588 | GFI1 | GFI1 | −0.42 | $9.1 \times 10^{-5}$ | −0.59 | $1.4 \times 10^{-7}$ | −0.03 | 0.80 |
| cg25648203 | 5 | 395444 | AHRR | AHRR | −0.28 | $1.2 \times 10^{-2}$ | −0.42 | $4.3 \times 10^{-4}$ | −0.06 | 0.61 |
| cg19089201 | 7 | 45002287 | MYO1G | MYO1G | 0.15 | $1.8 \times 10^{-1}$ | 0.27 | $2.6 \times 10^{-2}$ | 0.23 | 0.05 |

**Table 4.** Significantly associated transcripts in cis for CpGs that are differentially methylated in smoking discordant monozygotic twin pairs.

| CpG | Gene | Z score | p-value | FDR |
|---|---|---|---|---|
| cg25648203 | EXOC3 | −7.34 | 2.11e−13 | 0 |
| cg19089201 | RP4-647J21.1 | 5.55 | 2.84e−8 | 0 |
| cg05575921 | EXOC3 | −4.86 | 0.00000119 | 0.00039 |
| cg21161138 | EXOC3 | −3.82 | 0.000133 | 0.0254 |

proteins that interact with the nicotinic acetylcholine receptor, and the related enrichment of Kegg pathway dopaminergic neuron. Methylation levels at these CpGs might be reactive to nicotine exposure. Furthermore, the CpGs in *CACNA1D* are in proximity of a GWAS locus for smoking initiation, suggesting that this might be a locus that is not only reactive to smoking exposure, but may also contribute to smoking behaviour. Although it remains to be established if the epigenetic and genetic variation at this locus are functionally connected (i.e. have the same downstream consequences on gene expression), these results suggest that these CpGs can be interesting candidates for further studies into peripheral biomarkers of smoking addiction. Since we applied a discordant monozygotic twin design, the methylation differences identified at this locus in our study cannot be driven by mQTL effects of the SNPs associated with smoking. The data from monozygotic pairs discordant for former smoking indicate that methylation patterns are to a large extent reversible upon smoking cessation, which is in line with DNA methylation patterns being reactive to smoking. Nevertheless, our findings do not rule out that the possibility that reverse causation (DNA methylation driving smoking behaviour) might also contribute to the (maintenance of) smoking discordance in smoking discordant monozygotic twin pairs. Future analyses combining DNA methylation and genetic data from monozygotic and dizygotic twins may be applied to examine bidirectional causal associations between DNA methylation and smoking (*Minică et al., 2018*).

The main strength of our study is the use of the discordant monozygotic twin design to examine the effects of smoking, because it rules out genetic confounding, as well as many other confounding factors. The value of studying smoking effects against an identical genetic background is clear if one considers that one of the most strongly associated genetic variants for nicotine dependence is located in the DNA methyltransferase gene *DNMT3B* (*Hancock et al., 2018*). This strongly implies a role for DNA methylation in nicotine addiction, but it also suggests that horizontal genetic pleiotropy might contribute to associations between DNA methylation and smoking in ordinary case–control EWASs, where differences in DNA methylation between unrelated smokers and non-smokers may reflect differences in genotype. Our analysis had adequate power to detect large effects (i.e., the top hits identified in typical smoking EWAS) (*Tsai and Bell, 2015*). These reflect only a small proportion, however, of all smoking-associated sites. In our analysis of 53 monozygotic twin pairs discordant for current versus never smoking, we detected 13 CpGs at genome-wide significance, which represent 0.5% of the total number of CpGs (2623) detected in the smoking meta-analysis of unrelated individuals (2433 current verus 6956 never smokers) (*Joehanes et al., 2016*). The within-pair difference in smoking discordant monozygotic pairs was smaller compared to the effect size reported previously based on the comparison of unrelated smokers and non-smokers. Larger sample sizes are required to achieve adequate power to detect smaller effects. While the pattern of within-pair differences in current/never, current/former and former/never discordant monozygotic twin pairs was clearly in line with reversal of methylation patterns following smoking cessation, we did not find a strong correlation between within-pair differences in DNA methylation and time since quitting smoking in former smoking discordant pairs. If most reversal takes place gradually in the first view years after smoking cessation, it might require larger sample sizes of twin pairs discordant for recently quitting smoking to detect such a correlation. Larger samples sizes may be achieved by combining data from multiple twin cohorts in a meta-analysis.

Common limitations that apply to many EWA studies including ours are that we only analysed DNA methylation data from blood and that the technique used to measure DNA methylation only covers a small subset of all CpG sites in the genome. Another limitation is that information on smoking was obtained through self-report. We previously described smoking misclassification in this cohort based

on blood levels of cotinine (*van Dongen et al., 2018*), a biomarker for nicotine exposure, that has been measured in a subset of the cohort (*Bot et al., 2013*), which indicated a low misclassification rate. Plasma cotinine levels were available for 591 individuals classified as never smokers by self-report. Five of these individuals (0.8%) had cotinine levels ≥15 ng/ml, which is indicative of smoking, and thus indicates a misclassification of smoking status. In the current paper, we further showed that the correlation between cotinine levels in concordant current smoking pairs was similar to the correlation between self-reported number of cigarettes per day.

## Conclusion

In conclusion, we studied reactiveness of DNA methylation in blood cells to smoking and reversibility of methylation patterns upon quitting smoking in monozygotic twins. Analyses in special groups such as monozygotic twins are valuable to validate results from large population-based EWAS meta-analyses, or to train more accurate methylation scores for environmental exposures that are not confounded by genetic effects. Our results illustrate the potential to utilize DNA methylation profiles of monozygotic twins as a read out of discordant exposures at present and in the past.

## Acknowledgements

NTR warmly thanks all participants. We thank Conor Dolan for providing feedback on the manuscript. We acknowledge the contributions of the investigators of the BIOS consortium (*Supplementary file 9*): Bastiaan T Heijmans, Peter AC 't Hoen, Joyce van Meurs, Aaron Isaacs, Rick Jansen, Lude Franke, René Pool, Jouke J Hottenga, Marleen MJ van Greevenbroek, Coen DA Stehouwer, Carla JH van der Kallen, Casper G Schalkwijk, Cisca Wijmenga, Sasha Zhernakova, Ettje F Tigchelaar, P Eline Slagboom, Marian Beekman, Joris Deelen, Diana van Heemst, Jan H Veldink, Leonard H van den Berg, Cornelia M van Duijn, Bert A Hofman, Aaron Isaacs, André G Uitterlinden, P Mila Jhamai, Michael Verbiest, H Eka D Suchiman, Marijn Verkerk, Ruud van der Breggen, Jeroen van Rooij, Nico Lakenberg, Hailiang Mei, Maarten van Iterson, Michiel van Galen, Jan Bot, Dasha V Zhernakova, Rick Jansen, Peter van 't Hof, Patrick Deelen, Irene Nooren, Matthijs Moed, Martijn Vermaat, René Luijk, Marc Jan Bonder, Freerk van Dijk, Michiel van Galen, Wibowo Arindrarto, Szymon M Kielbasa, Morris A Swertz, and Erik W van Zwet.

## Additional information

### Group author details

#### BIOS Consortium

**Bastiaan T Heijmans**: Molecular Epidemiology Section, Department of Medical Statistics and Bioinformatics, Leiden University Medical Center, Leiden, Netherlands; **Peter AC 't Hoen**: Department of Human Genetics, Leiden University Medical Center, Leiden, Netherlands; **Joyce van Meurs**: Department of Internal Medicine, Erasmus MC, Rotterdam, Netherlands; **Aaron Isaacs**: Department of Genetic Epidemiology, Erasmus MC, Rotterdam, Netherlands; **Rick Jansen**: Department of Psychiatry, VU University Medical Center, Neuroscience Campus Amsterdam, Amsterdam, Netherlands; **Lude Franke**: Department of Genetics, University of Groningen, University Medical Centre Groningen, Groningen, Netherlands; **Dorret I Boomsma**: Department of Biological Psychology, VU University Amsterdam, Neuroscience Campus Amsterdam, Amsterdam, Netherlands; **René Pool**: Department of Biological Psychology, VU University Amsterdam, Neuroscience Campus Amsterdam, Amsterdam, Netherlands; **Jenny van Dongen**: Department of Biological Psychology, VU University Amsterdam, Neuroscience Campus Amsterdam, Amsterdam, Netherlands; **Jouke J Hottenga**: Department of Biological Psychology, VU University Amsterdam, Neuroscience Campus Amsterdam, Amsterdam, Netherlands; **Marleen MJ van Greevenbroek**: Department of Internal Medicine and School for Cardiovascular Diseases (CARIM), Maastricht University Medical Center, Maastricht, Netherlands; **Coen DA Stehouwer**: Department of Internal Medicine and School for Cardiovascular Diseases (CARIM), Maastricht University Medical Center, Maastricht, Netherlands; **Carla JH van der Kallen**: Department of Internal Medicine and School for Cardiovascular Diseases (CARIM), Maastricht University Medical Center, Maastricht, Netherlands; **Casper G Schalkwijk**: Department

of Internal Medicine and School for Cardiovascular Diseases (CARIM), Maastricht University Medical Center, Maastricht, Netherlands; **Cisca Wijmenga**: Department of Genetics, University of Groningen, University Medical Centre Groningen, Groningen, Netherlands; **Sasha Zhernakova**: Department of Genetics, University of Groningen, University Medical Centre Groningen, Groningen, Netherlands; **Ettje F Tigchelaar**: Department of Genetics, University of Groningen, University Medical Centre Groningen, Groningen, Netherlands; **P Eline Slagboom**: Molecular Epidemiology Section, Department of Medical Statistics and Bioinformatics, Leiden University Medical Center, Leiden, Netherlands; **Marian Beekman**: Molecular Epidemiology Section, Department of Medical Statistics and Bioinformatics, Leiden University Medical Center, Leiden, Netherlands; **Joris Deelen**: Molecular Epidemiology Section, Department of Medical Statistics and Bioinformatics, Leiden University Medical Center, Leiden, Netherlands; **Diana van Heemst**: Department of Gerontology and Geriatrics, Leiden University Medical Center, Leiden, Netherlands; **Jan H Veldink**: Department of Neurology, Brain Center Rudolf Magnus, University Medical Center Utrecht, Utrecht, Netherlands; **Leonard H van den Berg**: Department of Neurology, Brain Center Rudolf Magnus, University Medical Center Utrecht, Utrecht, Netherlands; **Cornelia M van Duijn**: Department of Genetic Epidemiology, Erasmus MC, Rotterdam, Netherlands; **Bert A Hofman**: Department of Epidemiology, Erasmus MC, Rotterdam, Netherlands; **André G Uitterlinden**: Department of Internal Medicine, Erasmus MC, Rotterdam, Netherlands; **P Mila Jhamai**: Department of Internal Medicine, Erasmus MC, Rotterdam, Netherlands; **Michael Verbiest**: Department of Internal Medicine, Erasmus MC, Rotterdam, Netherlands; **H Eka D Suchiman**: Molecular Epidemiology Section, Department of Medical Statistics and Bioinformatics, Leiden University Medical Center, Leiden, Netherlands; **Marijn Verkerk**: Department of Internal Medicine, Erasmus MC, Rotterdam, Netherlands; **Ruud van der Breggen**: Molecular Epidemiology Section, Department of Medical Statistics and Bioinformatics, Leiden University Medical Center, Leiden, Netherlands; **Jeroen van Rooij**: Department of Internal Medicine, Erasmus MC, Rotterdam, Netherlands; **Nico Lakenberg**: Molecular Epidemiology Section, Department of Medical Statistics and Bioinformatics, Leiden University Medical Center, Leiden, Netherlands; **Hailiang Mei**: Sequence Analysis Support Core, Leiden University Medical Center, Leiden, Netherlands; **Maarten van Iterson**: Molecular Epidemiology Section, Department of Medical Statistics and Bioinformatics, Leiden University Medical Center, Leiden, Netherlands; **Michiel van Galen**: Department of Human Genetics, Leiden University Medical Center, Leiden, Netherlands; **Jan Bot**: SURFsara, Amsterdam, Netherlands; **Dasha V Zhernakova**: Department of Genetics, University of Groningen, University Medical Centre Groningen, Groningen, Netherlands; **Peter van 't Hof**: Sequence Analysis Support Core, Leiden University Medical Center, Leiden, Netherlands; **Patrick Deelen**: Department of Genetics, University of Groningen, University Medical Centre Groningen, Groningen, Netherlands; **Irene Nooren**: SURFsara, Amsterdam, Netherlands; **Bastiaan T Heijmans**: Molecular Epidemiology Section, Department of Medical Statistics and Bioinformatics, Leiden University Medical Center, Leiden, Netherlands; **Matthijs Moed**: Molecular Epidemiology Section, Department of Medical Statistics and Bioinformatics, Leiden University Medical Center, Leiden, Netherlands; **Martijn Vermaat**: Department of Human Genetics, Leiden University Medical Center, Leiden, Netherlands; **René Luijk**: Molecular Epidemiology Section, Department of Medical Statistics and Bioinformatics, Leiden University Medical Center, Leiden, Netherlands; **Marc Jan Bonder**: Department of Genetics, University of Groningen, University Medical Centre Groningen, Groningen, Netherlands; **Freerk van Dijk**: Genomics Coordination Center, University Medical Center Groningen, University of Groningen, Groningen, Netherlands; **Wibowo Arindrarto**: Sequence Analysis Support Core, Leiden University Medical Center, Leiden, Netherlands; **Szymon M Kielbasa**: Medical Statistics Section, Department of Medical Statistics and Bioinformatics, Leiden University Medical Center, Leiden, Netherlands; **Morris A Swertz**: Genomics Coordination Center, University Medical Center Groningen, University of Groningen, Groningen, Netherlands; **Erik W van Zwet**: Medical Statistics Section, Department of Medical Statistics and Bioinformatics, Leiden University Medical Center, Leiden, Netherlands; **Peter-Bram 't Hoen**: Department of Human Genetics, Leiden University Medical Center, Leiden, Netherlands

**Competing interests**
BIOS Consortium: The other authors declare that no competing interests exist.

## Funding

| Funder | Grant reference number | Author |
|---|---|---|
| National Institute on Drug Abuse | DA049867 | Michael C Neale |
| ZonMw | NWO-Groot 480-15-001/674 | Gonneke Willemsen<br>Eco JC de Geus<br>Dorret I Boomsma |

The funders had no role in study design, data collection, and interpretation, or the decision to submit the work for publication.

## Author contributions

Jenny van Dongen, Conceptualization, Formal analysis, Writing - original draft; Gonneke Willemsen, Data curation, Funding acquisition, Writing - review and editing; BIOS Consortium, Data curation, Funding acquisition, Methodology, Software; Eco JC de Geus, Dorret I Boomsma, Michael C Neale, Supervision, Funding acquisition, Writing - review and editing

## Author ORCIDs

Jenny van Dongen http://orcid.org/0000-0003-2063-8741

## Ethics

Informed consent was obtained from all participants. The study was approved by the Central Ethics Committee on Research Involving Human Subjects of the VU University Medical Centre, Amsterdam, an Institutional Review Board certified by the U.S. Office of Human Research Protections (IRB number IRB00002991 under Federal-wide Assurance – FWA00017598; IRB/institute code, NTR 03-180).

## Decision letter and Author response

Decision letter https://doi.org/10.7554/eLife.83286.sa1
Author response https://doi.org/10.7554/eLife.83286.sa2

# Additional files

## Supplementary files

• Supplementary file 1. Questions on smoking that were asked at blood draw and quality control of longitudinal smoking data.

• Supplementary file 2. Genomic inflation factor of each epigenome-wide association study (EWAS) analysis.

• Supplementary file 3. Within-pair analysis results for 13 epigenome-wide significant CpGs in current/never discordant pairs.

• Supplementary file 4. Epigenome-wide significant CpGs in MZ twin pairs discordant for current/former smoking.

• Supplementary file 5. Correlations between the within-pair difference in DNA methylation level and time since quitting smoking in MZ pairs discordant for former smoking.

• Supplementary file 6. Epigenome-wide association study (EWAS) atlas trait enrichment analysis results for top CpGs identified in current/never smoking discordant twin pairs.

• Supplementary file 7. Kegg pathway enrichment analysis results for top CpGs identified in current/never smoking discordant twin pairs.

• Supplementary file 8. GO pathway enrichment analysis results for top CpGs identified in current/never smoking discordant twin pairs.

• Supplementary file 9. Biobank-based Integrative Omics Studies (BIOS) consortium investigators.

• MDAR checklist

• Source code 1. R-script for within-MZ pair epigenome-wide association study (EWAS analysis).

• Reporting standard 1. This manuscript follows the guidelines of STROBE (Strengthening the reporting of observational studies in epidemiology). The STROBE checklist has been included in the submission.

## Data availability

The HumanMethylation450 BeadChip data from the NTR are available as part of the Biobank-based Integrative Omics Studies (BIOS) Consortium in the European Genome-phenome Archive (EGA), under the accession code EGAD00010000887, https://ega-archive.org/datasets/EGAD00010000887 (Study ID EGAS00001001077, Title: The mission of the BIOS Consortium is to create a large-scale data infrastructure and to bring together BBMRI researchers focusing on integrative omics studies in Dutch Biobanks, contact: The BIOS Consortium: Biobank-based Integrative Omics Studies, Contact person: Rick Jansen). The OMICs data and additional phenotype data are available upon request via the BBMRI-NL BIOS consortium (https://www.bbmri.nl/acquisition-use-analyze/bios). All NTR data can be requested by bona fida researchers (https://ntr-data-request.psy.vu.nl/). Because of the consent given by study participants the data cannot be made publicly available. The pipeline for DNA methylation-array analysis developed by the Biobank-based Integrative Omics Study (BIOS) consortium is available here: https://molepi.github.io/DNAmArray_workflow/ copy archived at *Sinke, 2020* (https://doi.org/10.5281/zenodo.3355292). The code for the EWAS analysis in monozygotic twin pairs is included in *Source code 1*.

The following dataset was generated:

| Author(s) | Year | Dataset title | Dataset URL | Database and Identifier |
|---|---|---|---|---|
| BIOS consortium or Biobank-Based Integrative Omics Consortium | 2023 | RP3 Freeze 1 | https://ega-archive.org/datasets/EGAD00010000887 | European Genome-Phenome Archive, EGAD00010000887 |

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
