## [Editor Report]

This study presents valuable findings regarding how smoking can leave a lasting imprint on the human genome. The twin pairs study design is unique, and the methods applied by the authors are solid, providing an excellent starting point for large translational studies with rigorous laboratory approaches. This work will be of interest to geneticists and genetic epidemiologists.

---

## [Decision Letter]

**Decision letter after peer review:**

Thank you for submitting your article "Effects of smoking on genome-wide DNA methylation profiles: A study of discordant and concordant monozygotic twin pairs" for consideration by *eLife*. Your article has been reviewed by 3 peer reviewers, including Melinda Aldrich as Reviewing Editor and Reviewer #1, and the evaluation has been overseen by W Kimryn Rathmell as the Senior Editor. The following individuals involved in review of your submission have agreed to reveal their identity: Jeff Craig (Reviewer #2); Jaakko Kaprio (Reviewer #3).

Essential revisions:

Request from Reviewing Editor:

One of the references cited by the authors (Allione et al., Plos One 2015) cites two papers (Andreoli et al. and Marcon et al) that utilize a study population collected and phenotyped with support from the tobacco industry, specifically by British American Tobacco. We request the Allione reference be annotated to indicate the study utilized data that was collected with support from the tobacco industry.*Reviewer #1 (Recommendations for the authors):*

1. There are several typos throughout that need to be corrected (extra words inserted, for example: line 208 says "To the study the overlap…" this should instead read "To study the overlap…").

2. The EWAS analysis says that the linear regression analysis corrected for array row. It is unclear to this reviewer why one would correct for array row. Can this be clarified?

3. It is unclear whether the 5 significant CpGs that were identified using the discordant current-former twin pair design have been previously identified or if these are novel findings.

4. What type of correlation test was performed? Please add this detail.

5. Please write out the abbreviation for DMP the first time used in the manuscript.

6. Clarification regarding the time period that was asked about for smoking behaviors would be helpful. For example, the authors indicate current smokers were asked how many cigarettes per day they smoked, but no time period is provided. Same for former smokers. Also, this reviewer assumes the authors are referring to the average number of cigarettes were smoked per day. If the authors could clarify these points that would be useful for the reader.

*Reviewer #2 (Recommendations for the authors):*

1. Introduction, lines 111-112 "They [identical twins] have been exposed to similar prenatal conditions": please expand to differentiating between shared and nonshared intrauterine exposures.

2. Results, lines 228-9 "At four of the 13 CpGs, DNA methylation level in blood was associated with the expression level of nearby genes." Can you test whether this number would have been achieved by chance?

3. Results: could you compare average effect sizes with studies of singletons to test whether genetic identity attenuated any effects?

4. Results line 161 "Genome-wide test statistics were not inflated": could you please explain how you came to this conclusion using Figure 2C, especially for those without experience os using this metric. Please also note that Figure 2C is too small to read even when viewed on an A4 page.

*Reviewer #3 (Recommendations for the authors):*

1. In the introduction, a recent review could also be cited (Heikkinen A, Bollepalli S, Ollikainen M. The potential of DNA methylation as a biomarker for obesity and smoking. J Intern Med. 2022 Sep;292(3):390-408.. PMID: 35404524).

2. Line 111, It should be noted that in MZ twins matching on early environment is not perfect, especially the prenatal environment with placentation and chorionicity effects (Martin et al., 1997, PMID: 9398838).

3. Line 163-164 How consistent was the difference within pairs? On average the smoking twin had lower methylation, but was this the case in all pairs?

4. Line 170 The average time since cessation was 9 years, but what was the variation (SD) and range. Did the authors define a minimum duration of smoking cessation for the smoker to be considered as a true former smoker. Persons who report very recent quitting are often found to continue some level of smoking (based on biomarkers).

5. Line 190 -191 The intrapair correlations of amount (CPD) and total exposure (packyears) within twin pairs in which both smoke are consistent with prior literature. Could the author speculate on why the correlations are not higher – is there a role for measurement error, or differing brands of cigarettes being smoked but yielding equal amounts of nicotine (which is not measured). Would biomarkers have provided higher correlations?

6. Line 208 is a strange sentence that does not make sense.

In addition to the discordant pair analyses, I suggest they run bivariate twin analyses to evaluate shared genetics (a) of smoking with the methylation values of top CpGs, and (b) of smoking with a epigenome-wide predicted smoking score (such EpiSmokEr, PMIDs 35716602, 31466478). Such analyses would help to address to what extent within-pair analyses capture differences seen between individuals.

7. Line 237 I would like to see some more discussion and evidence for or against their second potential explanation of Reverse causation. Given the complex nature of smoking behavior, is it at all likely that methylation drives smoking behavior?

8. Line 295-297 Can the authors quantify what fraction of individual based EWAS hits are identified here and how much of the difference between smokers and never smokers is accounted for by the observed differences in twin pairs discordant for smoking.

9. Topics for discussion that could have been included.

a) How substantial are the effect of misclassification of smoking status and of amount smoked (see related comment on assessment of smoking behavior).

b) Metabolism of nicotine to cotinine and related metabolites. Cotinine is a well-known biomarker of recent tobacco use/nicotine exposure. Methylation associations with cotinine levels have been published (Gupta R et al., 2019. PMID: 30611298; Lee MK et al., 2016 PMID: 27688819), which specifically address the relationship between nicotine and methylation (rather than all exposures in tobacco). Recent model organism work could also be cited (Peng et al., 2022, PMID 36119846).

10. Methods: Line 338 – Are there any ethnicity effects? Please provide more detail on the pairs (in Table 1 or in text) on their socio-economic status, marital status and spousal tobacco use, and other behavioral traits that affect methylation (such as obesity, alcohol use, traumatic events). Did the birth weights and birth order of the smoking and non-smoking twins in a discordant pair differ?

11. Line 339 From how many families are the 3055 individuals? A comparison with DZ twins discordant for smoking would be a valuable addition, to tease apart effects of controlling for genetics and shared environment (in MZ pairs) versus some genetics and shared environment (in DZ pairs). The genetic risk for smoking could be controlled for in DZ pairs using polygenic risk scores for smoking behavior from the GSCAN consortium and/or UK Biobank.

12. Line 377 Smoking

Was the interview for smoking behavior more detailed than described here? How was a regular smoker defined? For example the given question "Did you ever smoke? (line 380) implies two answers: yes and no, and a yes answer is an ever smoker. It does not distinguish between current and former smokers. Please provide the actual questions used in a supplement, or a link to an appropriate webpage with the items (in Dutch and English).

13. Given that the participants have answered multiple surveys, can you document the consistency of the responses over time. For example, how many who now reported having never smokers had reported smoking in an earlier survey?

How did you handle non-daily smokers? Are they considered non-smokers, smokers or excluded?

14. Did you ask any pairs discordant for smoking why one had initiated smoking and the other did not., likewise why one quit and the other did not?

15. Are there any validation studies of smoking status, using biomarkers such as cotinine or carbon monoxide in the NTR?

16. Did smoking assess cigarette use only or all smoked tobacco products such as cigars and pipe use. What about smokeless tobacco/snus, e-cigs (rare at that time I believe) or nicotine replacement therapy as a source of nicotine?

17. Line 406 uses the wording " smoking discordant monozygotic twins". I would use twin pairs discordant for smoking OR smoking discordant pairs consistently as the pairs are discordant, not the individual twins.

18. Line 432 The Bonferroni correction is overly conservative here as the CpG sites are correlated, so that should be taken into account.

19. References: There is missing or erroneous information in the cited literature for example line 636 has no author names, lines 555, 562, 608, 633, 643, 576, 586, 605 etc no issue and/or page info, etc. No publisher (line 621) Please check all references.

[Editors’ note: further revisions were suggested prior to acceptance, as described below.]

Thank you for resubmitting your work entitled "Effects of smoking on genome-wide DNA methylation profiles: A study of discordant and concordant monozygotic twin pairs" for further consideration by *eLife*. Your revised article has been evaluated by W Kimryn Rathmell (Senior Editor) and a Reviewing Editor.

The manuscript has been improved but there are some remaining issues that need to be addressed, as outlined below:

– In their rebuttal, the authors mention cigs per day was measured at the present time for current smokers and for former smokers no time period was requested. This leaves it vague as to the time period the former smoker respondents provided, e.g. was it on average over the time smoked or during the last year they smoked? A brief acknowledgment should be included that the time period was not captured when assessing cigs per day for former smokers. Thus, it was left to the respondent to determine the time period, which could lead to variation in reporting by respondents.

– The authors corrected smoking status using longitudinal survey data but did not indicate that they made these corrections in the revised manuscript, only in the rebuttal. This detail should be included for transparency. Related to this, the authors provide in their response and revised manuscript the cotinine levels for a large subset of the never smoking participants. For most, the cotinine levels were consistent with amounts expected for never smokers, but there were 5 (0.8%) persons that had cotinine levels indicative of a current smoker. Can the authors confirm corrections to smoking status were not made with cotinine? Or if they were corrected, this should be mentioned in the manuscript.

– Line 574 has a typo: ‘low classification rate’ should read ‘low misclassification rate’.

– In the initial review, one of the reviewers asked for information about the statistical test for inflation of the GWAS results. The authors indicate in their response that a sentence was added to the manuscript about this inflation factor, but it appears this sentence may have been mistakenly omitted from the manuscript.

– In the prior review, it was requested the authors remove the phrase "smoking discordant monozygotic twins" and instead refer to pairs. They made the requested revision, but then added it back into one of their revised sentences (line 369). This should be adjusted to address the reviewer's comment.

---

## [Author Response]

Essential revisions:Request from Reviewing Editor:One of the references cited by the authors (Allione et al. Plos One 2015) cites two papers (Andreoli et al. and Marcon et al) that utilize a study population collected and phenotyped with support from the tobacco industry, specifically by British American Tobacco. We request the Allione reference be annotated to indicate the study utilized data that was collected with support from the tobacco industry.

We thank the editor for this comment and have added this information to the reference list.

Reviewer #1 (Recommendations for the authors):1. There are several typos throughout that need to be corrected (extra words inserted, for example: line 208 says "To the study the overlap…" this should instead read "To study the overlap…").

We thank the reviewer for the careful reading of our manuscript. We thoroughly checked the manuscript and corrected typos.

2. The EWAS analysis says that the linear regression analysis corrected for array row. It is unclear to this reviewer why one would correct for array row. Can this be clarified?

Each Illumina 450k array contains 6 rows and 2 columns to fit 12 samples in total. Array row is a technical confounder with a linear effect on DNA methylation signals. This is connected to the technical procedure of fluorescent staining in which the arrays are placed in vertical orientation in the machine and fluorescent dye is injected at the top, resulting in a signal gradient across samples from the top to the bottom row of the array. Most of this effect (at the global level) is removed with normalization, however, probe-specific effects typically persist. Probe-specific effects of array row can be corrected for efficiently by including array row number as a covariate. It is common practice in Illumina DNA methylation array analysis to correct for array row (see for example the following reference on commonly used analysis strategies for epigenome-wide association studies https://doi.org/10.1186/s13059-019-1878-x; we have now cited this paper in our methods section).

3. It is unclear whether the 5 significant CpGs that were identified using the discordant current-former twin pair design have been previously identified or if these are novel findings.

Thank you for pointing this out. We have now clarified that all CpGs were previously identified:

“All 13 differentially methylated CpGs identified in current smoking-discordant pairs and all 10 CpGs identified in former-smoking discordant pairs have been previously associated with smoking.”

4. What type of correlation test was performed? Please add this detail.

We’ve now clarified in the methods section that Pearson correlations were used.

5. Please write out the abbreviation for DMP the first time used in the manuscript.

We did this upon first occurrence (results, page 7, line 162).

6. Clarification regarding the time period that was asked about for smoking behaviors would be helpful. For example, the authors indicate current smokers were asked how many cigarettes per day they smoked, but no time period is provided. Same for former smokers. Also, this reviewer assumes the authors are referring to the average number of cigarettes were smoked per day. If the authors could clarify these points that would be useful for the reader.

We apologize that this was unclear. During the home visit, blood samples were collected and participants were interviewed. Thus, at blood draw, current smokers were asked how many cigarettes they smoked per day at present (at the moment of blood draw), with the simple question: “how many cigarettes do you smoke per day?” and former smokers were asked how many cigarettes per day they used to smoke (“how many cigarettes did you use to smoke per day?”). We’ve edited the following sentences to clarify this.

Page 15, line 383: “Current smokers were asked how many years they smoked and how many cigarettes per day they smoked at present, while ex-smokers were asked how many years ago they quit, for how many years they smoked and how many cigarettes per day they smoked.”

Reviewer #2 (Recommendations for the authors):1. Introduction, lines 111-112 "They [identical twins] have been exposed to similar prenatal conditions": please expand to differentiating between shared and nonshared intrauterine exposures.

Thank you for this suggestion. We have modified this section as follows:

“On the other hand, monozygotic twins are genetically identical (except for de novo mutations, but these are rare), share a womb, and are matched on sex, age and childhood environment. They have been exposed to similar prenatal conditions, which may include second hand smoke from smoking mothers and others. Differences in prenatal environment of monozygotic twins due to for instance unequal vascular supply are also recognized, although it remains to be investigated to what extent the impact of prenatal smoke exposure might differ between monozygotic twins.”

2. Results, lines 228-9 "At four of the 13 CpGs, DNA methylation level in blood was associated with the expression level of nearby genes." Can you test whether this number would have been achieved by chance?

The associations between DNA methylation and gene expression are unlikely due to chance, because multiple testing correction was applied. Associations between DNA methylation and gene expression were previously tested in an independent whole blood RNA-sequencing dataset from the Biobank-based Integrative Omics Study (BIOS) consortium that did not include NTR, and which tested associations between all genome-wide CpGs and transcripts in cis (<250 kb). In this previous study, we looked up for which of our top CpGs the methylation levels were significantly associated with gene expression levels at the experiment-wide threshold applied by this study (FDR<5.0%, across all genome-wide CpGs and transcripts *in cis*). Hence these associations are unlikely due to chance.

3. Results: could you compare average effect sizes with studies of singletons to test whether genetic identity attenuated any effects?

We have added a comparison of the effect sizes observed in the discordant monozygotic twin pairs to the effect sizes observed previously in a large EWAS meta-analysis of unrelated smokers and non-smokers to the Results section.

Page 7, line 164: “At 11 of the 13 CpGs, the methylation difference in smoking discordant monozygotic twin pairs was smaller (on average 19.0%, range=2.2-37.5%) compared to the methylation difference reported previously in an EWAS meta-analysis of smoking. At two CpGs, the methylation difference in smoking discordant monozygotic twins was larger (on average 24.6%).”

4. Results line 161 "Genome-wide test statistics were not inflated": could you please explain how you came to this conclusion using Figure 2C, especially for those without experience os using this metric. Please also note that Figure 2C is too small to read even when viewed on an A4 page.

Please note that this cannot be directly assessed based on figure 2C, because in this figure, 2 sets of p-values are plotted (this figure is meant to illustrate the p-value distribution of CpGs inside and outside of smoking-associated genetic regions identified in GWAS). The sentence quoted refers to Additional file 1, which gives the λ (inflation factor) for each analysis (all were close to 1, indicating no inflation of test statistics). We realized that we had omitted to describe how we assessed inflation, and have added the following sentence to the methods section. We also improved the readability of the figures.

“For each EWAS analysis, the R package Bacon was used to compute the Bayesian inflation factor.”

Reviewer #3 (Recommendations for the authors):1. In the introduction, a recent review could also be cited (Heikkinen A, Bollepalli S, Ollikainen M. The potential of DNA methylation as a biomarker for obesity and smoking. J Intern Med. 2022 Sep;292(3):390-408.. PMID: 35404524).

Thank you for sharing the reference, we’ve added it to the introduction.

2. Line 111, It should be noted that in MZ twins matching on early environment is not perfect, especially the prenatal environment with placentation and chorionicity effects (Martin et al., 1997, PMID: 9398838).

Thank you for this suggestion. We have modified this section as follows:

“On the other hand, monozygotic twins are genetically identical (except for de novo mutations, but these are rare), share a womb, and are matched on sex, age and childhood environment. They have been exposed to similar prenatal conditions, which may include second hand smoke from smoking mothers and others. Differences in prenatal environment of monozygotic twins due to for instance unequal vascular supply are also recognized, although it remains to be investigated to what extent the impact of prenatal smoke exposure might differ between monozygotic twins.”

3. Line 163-164 How consistent was the difference within pairs? On average the smoking twin had lower methylation, but was this the case in all pairs?

The differences were highly consistent. We’ve added figures of the raw methylation β-values of the discordant twin pairs for the 13 top CpGs (Figure 2 —figure supplement 1), and added the following sentence to the Results section:

“Pair-level methylation β-values are show in Figure 2 —figure supplement 1 and illustrate large consistency in the direction of effect. For example, at top CpG site cg05575921, for 51 out of the 53 discordant pairs, the smoking twin had a lower methylation level than the non-smoking twin.”

4. Line 170 The average time since cessation was 9 years, but what was the variation (SD) and range. Did the authors define a minimum duration of smoking cessation for the smoker to be considered as a true former smoker. Persons who report very recent quitting are often found to continue some level of smoking (based on biomarkers).

Thank you for this suggestion. We’ve added SD and range of quitting smoking. We did not apply a threshold for minimum duration of reported quitting smoking. We acknowledge that a small percentage of smoking statuses may be misclassified. Plasma cotinine level (which we have now added to table 1), indeed suggest active cigarette smoking for a small number of individuals whose self-reported status is former-smoker, while cotinine levels are essentially consistent with smoking status in the other groups (current and never smokers). We have now added the topic of misclassification to the discussion.

“In twin pairs discordant for former smoking (N=72 pairs, mean age=41 years), the twins, who used to smoke, had quit smoking on average 14 years ago (SD=11.4,range=0.04-50 years), while the other twins had never initiated regular smoking”

“By contrast, in twin pairs of which one twin was a current smoker at blood draw and the co-twin had quit smoking (on average 9 years, ago, SD=10.2, range=0.02-40 years, N=66 pairs),”

“Another limitation is that information on smoking was obtained through self-report. We previously described smoking misclassification in this cohort based on blood levels of cotinine, a biomarker for nicotine exposure, that has been measured in a subset of the cohort, which indicated a low classification rate. Plasma cotinine levels were available for 591 individuals classified as never smokers by self-report. Five of these individuals (0.8%) had cotinine levels > = 15 ng/mL, which is indicative of smoking, and thus indicates a misclassification of smoking status. In the current paper, we further showed that the correlation between cotinine levels in concordant current smoking pairs was similar to the correlation between self-reported number of cigarettes per day.”

5. Line 190 -191 The intrapair correlations of amount (CPD) and total exposure (packyears) within twin pairs in which both smoke are consistent with prior literature. Could the author speculate on why the correlations are not higher – is there a role for measurement error, or differing brands of cigarettes being smoked but yielding equal amounts of nicotine (which is not measured). Would biomarkers have provided higher correlations?

This is an interesting question. We’ve now also looked at the correlation between plasma levels of cotinine in the smoking-concordant twin pairs, and the correlation (r=0.58) is only slightly higher compared to the correlation for self-reported cigarettes per day (r=0.50), and the correlation for packyears (r=0.43), i.e. the biomarker level is quite consistent with the similarity in reported amounts of cigarettes smoked per day. One could argue that the exact amount (or brand) smoked is not only subject to genetic influences but is an individual habit that is also influenced by the unique environment, individual life trajectories of the twins and potentially by differences in nicotine metabolic rate. An alternative explanation could be that the twins are in fact typically smoking similar amounts, but that self-report and measured cotinine levels have similar amounts of measurement error.

“The twin correlations in current smoking monozygotic twin pairs were *r*=0.50, p=2.2x10^-6^ for cigarettes per day (Figure 3b), *r*=0.43, p=3.2 x10^-4^ for packyears, and *r*=0.58, p=1.6x10^-8^, for plasma cotinine levels, respectively.”

6. Line 208 is a strange sentence that does not make sense.

We have adjusted the sentence.

“To study overlap of EWAS signal with genetic findings for smoking, we compared our EWAS results against GWAS results from the largest GWAS meta-analysis of smoking phenotypes. This is the meta-analysis of smoking initiation by the GWAS and Sequencing Consortium of Alcohol and Nicotine use (GSCAN)”

In addition to the discordant pair analyses, I suggest they run bivariate twin analyses to evaluate shared genetics (a) of smoking with the methylation values of top CpGs, and (b) of smoking with a epigenome-wide predicted smoking score (such EpiSmokEr, PMIDs 35716602, 31466478). Such analyses would help to address to what extent within-pair analyses capture differences seen between individuals.

We agree that this is a great suggestion for follow-up, that we are in fact currently working on (van Dongen et al. 2021 BGA Conference Abstract). This work will be included in a next manuscript. In the current manuscript, we choose to focus on the discordant/concordant monozygotic twin analysis. At present, there is only one previous study on DNA methylation in smoking discordant monozygotic twins.

van Dongen, Jenny, et al. "Examining Causality of the Association Between Smoking and DNA Methylation." *Behavior Genetics* 51.6 (2021): 749-750.

7. Line 237 I would like to see some more discussion and evidence for or against their second potential explanation of Reverse causation. Given the complex nature of smoking behavior, is it at all likely that methylation drives smoking behavior?

Thank you for this suggestion, but there is not much more we can say about it based on the current results. As already mentioned in the discussion, it is not unlikely that DNA methylation in relevant brain regions contributes to smoking behavior, however there is currently no evidence that DNA methylation in blood cells may reflect causal effects on smoking behaviour.

Discussion line 271 “Importantly, effects of smoking on DNA methylation in brain cells have been hypothesized to contribute to addiction, but it is largely unknown to what extent addiction-related DNA methylation dynamics are captured in other tissues such as blood. Nicotinic receptors are especially abundant in the central and peripheral nervous system, but are also present in other tissues. In peripheral blood, nicotinic receptors are present on lymphocytes and polymorphonuclear cells, suggesting that EWA studies performed on blood cells might capture nicotine-reactive methylation patterns.”

We have also added the following sentences to the discussion:

“The data from monozygotic pairs discordant for former smoking indicate that methylation patterns are to a large extent reversible upon smoking cessation, which is in line with DNA methylation patterns being reactive to smoking. Nevertheless, our findings do not rule out the possibility that reverse causation (DNA methylation driving smoking behaviour) might contribute to the (maintenance of) smoking discordance in smoking discordant monozygotic twin pairs. Future analyses combining DNA methylation and genetic data from monozygotic and dizygotic twins may be applied to examine bi-directional causal associations between DNA methylation and smoking (Minică et al., 2018).Future analyses combining DNA methylation and genetic data from monozygotic and dizygotic twins may be applied to examine bi-directional causal associations between DNA methylation and smoking (Minica et al. 2018).”

8. Line 295-297 Can the authors quantify what fraction of individual based EWAS hits are identified here and how much of the difference between smokers and never smokers is accounted for by the observed differences in twin pairs discordant for smoking.

Thank you for this suggestion. We’ve added the following sentences to the discussion:

“These reflect only a small proportion, however, of all smoking-associated sites. In our analysis of 53 monozygotic twin pairs discordant for current versus never smoking, we detected 13 CpGs at genome-wide significance, which represent 0.5% of the total number of CpGs (2623) detected in the smoking meta-analysis of unrelated individuals (2433 current verus 6956 never smokers). The within-pair difference in smoking discordant monozygotic pairs was smaller compared to the effect size reported previously based on the comparison of unrelated smokers and non-smokers.”

We’ve also added the following to the results (see reply to reviewer 2, comment 3).

Page 7, line 164: “At 11 of the 13 CpGs, the methylation difference in smoking discordant monozygotic twin pairs was smaller (on average 19.0%, range=2.2-37.5%) compared to the methylation difference reported previously in an EWAS meta-analysis of smoking. At two CpGs, the methylation difference in smoking discordant monozygotic twins was larger (on average 24.6%).”

9. Topics for discussion that could have been included.a) How substantial are the effect of misclassification of smoking status and of amount smoked (see related comment on assessment of smoking behavior).

Thank you for this suggestion. We’ve added the following to the discussion.

“Another limitation is that information on smoking was obtained through self-report. We previously described smoking misclassification in this cohort based on blood levels of cotinine, a biomarker for nicotine exposure, that has been measured in a subset of the cohort, which indicated a low classification rate. Plasma cotinine levels were available for 591 individuals classified as never smokers by self-report. Five of these individuals (0.8%) had cotinine levels > = 15 ng/mL, which is indicative of smoking, and thus indicates a misclassification of smoking status. In the current paper, we further showed that the correlation between cotinine levels in concordant current smoking pairs was similar to the correlation between self-reported number of cigarettes per day.”

b) Metabolism of nicotine to cotinine and related metabolites. Cotinine is a well-known biomarker of recent tobacco use/nicotine exposure. Methylation associations with cotinine levels have been published (Gupta R et al., 2019. PMID: 30611298; Lee MK et al., 2016 PMID: 27688819), which specifically address the relationship between nicotine and methylation (rather than all exposures in tobacco). Recent model organism work could also be cited (Peng et al., 2022, PMID 36119846).

This is a great suggestion, thank you for pointing out these references. We’ve added the following sentences to the discussion.

“Previous EWAS studies based on blood cotinine levels, as a biomarker for nicotine exposure, and based on a polygenic scores for nicotine metabolism, reported differentially methylated CpGs that largely overlap with CpGs found in EWAS of smoking status. Furthermore, E-cigarette based nicotine exposure of mice has been shown to cause DNA methylation changes in white blood cells.”

10. Methods: Line 338 – Are there any ethnicity effects? Please provide more detail on the pairs (in Table 1 or in text) on their socio-economic status, marital status and spousal tobacco use, and other behavioral traits that affect methylation (such as obesity, alcohol use, traumatic events). Did the birth weights and birth order of the smoking and non-smoking twins in a discordant pair differ?

Thank you for the suggestion. We’ve added the most widely available information: BMI and educational attainment as a measure of SES (note that alcohol use was not measured at the time of blood sampling, and that spousal information on smoking is also not always available). This showed a significant difference in BMI in pairs discordant for current/former smoking (higher BMI in the former smoking twin), and no significant differences in educational attainment that would survive multiple testing correction. Note that these twins took part in the Adult Netherlands Twin Register, and have not been followed since birth. Therefore, we have limited information related to birth for this group. In addition, we’ve added information on ancestry of the cohort in the methods section. A strength of the within-MZ pair design is that it is robust to effects of genetic variation and ancestry.

“The twin pairs were primarily of Dutch-European ancestry. For 753 of the 768 MZ pairs who are included in the current study, ancestry could be derived from principal components (PCs) calculated from genome-wide SNP array data that were available for the twins (750 pairs) or for both of their parents (3 pairs). According to the genotype data PCs, 4.5% of the pairs classify as ancestry outliers.”

11. Line 339 From how many families are the 3055 individuals? A comparison with DZ twins discordant for smoking would be a valuable addition, to tease apart effects of controlling for genetics and shared environment (in MZ pairs) versus some genetics and shared environment (in DZ pairs). The genetic risk for smoking could be controlled for in DZ pairs using polygenic risk scores for smoking behavior from the GSCAN consortium and/or UK Biobank.

As indicated in our response to comment 6 and 7, we are currently performing follow-up work that will be included in a next manuscript. In the current manuscript, we deliberately choose to focus on the data from monozygotic twins only, to fully showcase the value of data from monozygotic twins. We believe the current analysis of ‘just monozygotic twins’ is already a valuable addition to the existing literature because of the uniqueness and strength of the design and because (to our surprise) only one earlier EWAS study of smoking discordant monozygotic twins has been published so far (which had a much smaller sample size than ours and only included current/never discordant pairs).

12. Line 377 SmokingWas the interview for smoking behavior more detailed than described here? How was a regular smoker defined? For example the given question "Did you ever smoke? (line 380) implies two answers: yes and no, and a yes answer is an ever smoker. It does not distinguish between current and former smokers. Please provide the actual questions used in a supplement or a link to an appropriate webpage with the items (in Dutch and English).

Thank you for this suggestion. To clarify, we have now added an English translation of the questions that were asked at blood draw to the supplement (additional file 8):

Additional file 8: Questions on smoking that were asked at blood draw

“Did you ever smoke?”

(1) no, I never smoked

(2) I’m a former smoker

2a How many years ago did you quit?

2b For how many years have you smoked?

2c How many cigarettes did you use to smoke per day?

(3) yes.

3a How many years have you smoked?

3b How many cigarettes do you smoke per day?

13. Given that the participants have answered multiple surveys, can you document the consistency of the responses over time. For example, how many who now reported having never smokers had reported smoking in an earlier survey?How did you handle non-daily smokers? Are they considered non-smokers, smokers or excluded?

We indeed have used the longitudinal surveys to compare (and correct, if necessary) the smoking status at blood draw with smoking status from longitudinal NTR. In case of in consistencies, smoking status has been adjusted. Out of 9628 participants of the NTR biobank 1 project, smoking status at blood draw was consistent with longitudinal surveys for 97.1% of participants. The remaining 2.9% included the following cases: For 0.3% the status at blood draw has been adjusted based on checks against longitudinal surveys (e.g. the person reported to have never smoked at blood draw, while they reported smoking in longitudinal surveys). For 2%, blood status at blood draw was missing, and has been added from survey data. For 0.1%, smoking status was set to missing due to unresolvable inconsistencies (such participants are not included in our study because of missing smoking status). For 0.4%, the status at blood draw was missing and it could also not be retrieved from surveys due to insufficient information(these participants are also not included in our study). This is summarized in Author response table 1.

**Author response table 1. sa2table1:** 

check					
	Frequency	Percent	Valid Percent	Cumulative Percent	
Valid	1,00 no changes in original status	9349	97.1	97.1	97.1
	2,00 original status were adjusted based on data checks	33	.3	.3	97.4
	3,00 original status was missing, added from survey data	192	2.0	2.0	99.4
	4,00 made missing based on data checks	11	.1	.1	99.6
	5,00 original status missing and insufficient data to add status	43	.4	.4	100.0
	Total	9628	100.0	100.0	

Note, we also looked up specifically the MZ pairs discordant for current/never smoking. For all of these twins, smoking status obtained at blood draw was consistent with longitudinal surveys.

When individuals reported that they smoked regularly they were classified as a smoker, but there was no cutoff on the number of times an individual smoked. Based on the reported number of cigarettes per day in current smokers, we can see that the majority report smoking at least one cigarette per day, but a few report less than one cigarette per day, indicative of non-daily smoking.

14. Did you ask any pairs discordant for smoking why one had initiated smoking and the other did not., likewise why one quit and the other did not?

We did not, unfortunately. That would be interesting information to obtain indeed.

15. Are there any validation studies of smoking status, using biomarkers such as cotinine or carbon monoxide in the NTR?

Yes, plasma cotinine levels were measured in a subset of the samples (4099 NTR participants, described in Bot et al. 2013 https://doi.org/10.1016/j.jpsychores.2013.08.016). We previously described smoking misclassification among individuals with DNA methylation data and cotinine levels in van Dongen et al. 2018 (https://doi.org/10.1038/s41539-018-0020-2): “Plasma cotinine levels were available for 591 individuals classified as never smokers by self-report. Five of these individuals (0.8%) had cotinine levels > = 15 ng/mL, which is indicative of smoking, and thus indicates a misclassification of smoking status”.

Bot, M. et al. Exposure to secondhand smoke and depression and anxiety: A report from two studies in the Netherlands. J. Psychosom. Res. 75, 431–436 (2013).

van Dongen, J., Bonder, M.J., Dekkers, K.F. *et al.* DNA methylation signatures of educational attainment. *npj Science Learn* 3, 7 (2018). https://doi.org/10.1038/s41539-018-0020-2

16. Did smoking assess cigarette use only or all smoked tobacco products such as cigars and pipe use. What about smokeless tobacco/snus, e-cigs (rare at that time I believe) or nicotine replacement therapy as a source of nicotine?

We focused on cigarette smoking; we did not include questions on other ways to take in nicotine (like e-cigarettes or water pipe) or on cannabis use at the time of blood draw. Note that snus use is not common in the Netherlands (in particular not at the time when blood sampling was carried out) and that E-cigarettes have also increased in popularity only more recently. Furthermore, a survey on E-cigarette use in the Dutch population that was conducted in the time period when our study took place pointed out that individuals who reported the use of E-cigarettes primarily consisted of individuals who also smoked conventional cigarettes (Willemsen et al. De elektronische sigaret. Gebruik, gezondheidsrisico’s, en effectiviteit als stopmethode. Ned Tijdschr Geneeskd. 2015;159:A9259.).

17. Line 406 uses the wording " smoking discordant monozygotic twins". I would use twin pairs discordant for smoking OR smoking discordant pairs consistently as the pairs are discordant, not the individual twins.

Thank you for the suggestion. We have corrected this.

18. Line 432 The Bonferroni correction is overly conservative here as the CpG sites are correlated, so that should be taken into account.

We are aware that Bonferroni correction is stringent, and have now acknowledged this in the methods section:

“Statistical significance was assessed following stringent Bonferroni correction for the number of methylation sites tested (α = 0.05/411,169 = 1.2 x 10^-7^).”

Both Bonferroni correction and FDR are commonly applied in EWA studies. In our experience, Bonferroni correction, although perhaps slightly too stringent, produces the most robust results. For example, we previously reported that the replication rate across cohorts is much higher for Bonferroni-significant CpGs than it is for CpGs that meet FDR 5% (https://doi.org/10.1186/s13059-019-1878-x).

19. References: There is missing or erroneous information in the cited literature for example line 636 has no author names, lines 555, 562, 608, 633, 643, 576, 586, 605 etc no issue and/or page info, etc. No publisher (line 621) Please check all references.

Thank you for pointing this out. We have carefully checked and corrected the reference list.

[Editors’ note: what follows is the authors’ response to the second round of review.]The manuscript has been improved but there are some remaining issues that need to be addressed, as outlined below:– In their rebuttal, the authors mention cigs per day was measured at the present time for current smokers and for former smokers no time period was requested. This leaves it vague as to the time period the former smoker respondents provided, e.g. was it on average over the time smoked or during the last year they smoked? A brief acknowledgment should be included that the time period was not captured when assessing cigs per day for former smokers. Thus, it was left to the respondent to determine the time period, which could lead to variation in reporting by respondents.

Indeed, we asked participants the simple question “how many cigarettes did you use to smoke”. We have added a sentence on the inherit limitations of this question:

“Current smokers were asked how many years they smoked and how many cigarettes per day they smoked at present, while ex-smokers were asked how many years ago they quit, for how many years they smoked and how many cigarettes per day they smoked (note that the question on cigarettes per day to former smoker did not specify a particular time period, which may introduce variation in responses).”

– The authors corrected smoking status using longitudinal survey data but did not indicate that they made these corrections in the revised manuscript, only in the rebuttal. This detail should be included for transparency.

Please note that we did not feel that additions were necessary during revision because it was already stated in the first submission of the manuscript:

Page 9 “Data were checked for consistencies and missing data were completed by linking this information to data from surveys filled out close to the time of biobanking within the longitudinal survey study of the NTR.”

We had now added additional information that we included in the rebuttal letter to the supplementary information:

“Out of 9628 participants of the NTR biobank 1 project, smoking status at blood draw was consistent with longitudinal surveys for 97.1% of participants. The remaining 2.9% included the following cases: For 0.3% the status at blood draw has been adjusted based on checks against longitudinal surveys (e.g. the person reported to have never smoked at blood draw, while they reported smoking in longitudinal surveys). For 2%, blood status at blood draw was missing, and has been added from survey data. For 0.1%, smoking status was set to missing due to unresolvable inconsistencies (such participants are not included in our study because of missing smoking status). For 0.4%, the status at blood draw was missing and it could also not be retrieved from surveys due to insufficient information (these participants are also not included in our study).”

And refer to this on page 9 of the main text: “More details on these checks are described in Supplementary file 1.”

Related to this, the authors provide in their response and revised manuscript the cotinine levels for a large subset of the never smoking participants. For most, the cotinine levels were consistent with amounts expected for never smokers, but there were 5 (0.8%) persons that had cotinine levels indicative of a current smoker. Can the authors confirm corrections to smoking status were not made with cotinine? Or if they were corrected, this should be mentioned in the manuscript.

No, self-reported smoking data were not adjusted based on cotinine levels.

– Line 574 has a typo: ‘low classification rate’ should read ‘low misclassification rate’.

Thank you. We have corrected this.

– In the initial review, one of the reviewers asked for information about the statistical test for inflation of the GWAS results. The authors indicate in their response that a sentence was added to the manuscript about this inflation factor, but it appears this sentence may have been mistakenly omitted from the manuscript.

We apologize for the omission. The sentence has now been added to page 11.

– In the prior review, it was requested the authors remove the phrase "smoking discordant monozygotic twins" and instead refer to pairs. They made the requested revision, but then added it back into one of their revised sentences (line 369). This should be adjusted to address the reviewer's comment.

We apologize for this mistake. We now use discordant pairs throughout the manuscript.